# Improved systemic AAV gene therapy with a neurotrophic capsid in Niemann–Pick disease type C1 mice

Cristin D Davidson[1],*, Alana L Gibson[1],*, Tansy Gu[1],*, Laura L Baxter[1], Benjamin E Deverman[2], Keith Beadle[2], Arturo A Incao[1], Jorge L Rodriguez-Gil[1], Hideji Fujiwara[3], Xuntian Jiang[3], Randy J Chandler[4], Daniel S Ory[3], Viviana Gradinaru[2], Charles P Venditti[4], William J Pavan[1]

Niemann–Pick C1 disease (NPC1) is a rare, fatal neurodegenerative disease caused by mutations in *NPC1*, which encodes the lysosomal cholesterol transport protein NPC1. Disease pathology involves lysosomal accumulation of cholesterol and lipids, leading to neurological and visceral complications. Targeting the central nervous system (CNS) from systemic circulation complicates treatment of neurological diseases with gene transfer techniques. Selected and engineered capsids, for example, adeno-associated virus (AAV)-PHP.B facilitate peripheral-to-CNS transfer and hence greater CNS transduction than parental predecessors. We report that systemic delivery to $Npc1^{m1N/m1N}$ mice using an AAV-PHP.B vector ubiquitously expressing *NPC1* led to greater disease amelioration than an otherwise identical AAV9 vector. In addition, viral copy number and biodistribution of GFP-expressing reporters showed that AAV-PHP.B achieved more efficient, albeit variable, CNS transduction than AAV9 in $Npc1^{m1N/m1N}$ mice. This variability was associated with segregation of two alleles of the putative AAV-PHP.B receptor *Ly6a* in $Npc1^{m1N/m1N}$ mice. Our data suggest that robust improvements in NPC1 disease phenotypes occur even with modest CNS transduction and that improved neurotrophic capsids have the potential for superior NPC1 AAV gene therapy vectors.

## Introduction

Niemann–Pick disease, type C (NPC) is a fatal, autosomal recessive lysosomal storage disorder with an estimated incidence of 1 in ~100,000 live births (Vanier, 2013). Unesterified cholesterol and sphingolipid accumulation in the lysosome is a primary hallmark of NPC. In 95% of NPC patients, mutations in *NPC1* (NPC1 disease, OMIM #257220), which encodes the NPC1 transmembrane protein found in the limiting lysosomal membrane (Carstea et al, 1997), is causative. The remaining 5% of patients have mutations in the soluble lysosomal protein *NPC2* (NPC2 disease, OMIM #607625), which binds cholesterol and physically interacts with NPC1 (Naureckiene et al, 2000). The two forms of the disease are clinically indistinguishable, consistent with the many studies demonstrating that NPC1 and NPC2 work together to regulate cholesterol efflux from the lysosome (Infante et al, 2008; Cologna & Rosenhouse-Dantsker, 2019; Pfeffer, 2019). NPC patients exhibit a wide array of neurological symptoms, including motor impairment and learning deficits, as well as visceral complications such as hepatosplenomegaly, with a highly heterogeneous disease severity and age of onset (Garver et al, 2007; Vanier, 2010; Patterson et al, 2013; Geberhiwot et al, 2018). Currently there are no FDA-approved therapies for NPC in the United States, thus there is an urgent need for discovery of effective treatments for this debilitating, fatal disease.

Gene therapy represents a promising treatment for monogenic diseases such as NPC1 and related lysosomal diseases. Recent technological advances such as improved vector design, RNA-based therapies, and CRISPR/Cas9 technology have brought gene therapy for these disorders closer to reality (Ma et al, 2019; Shahryari et al, 2019). In particular, engineered capsids derived from various serotypes of adeno-associated virus (AAV) have been found to be highly efficient for gene delivery and exhibit a wide range of tissue specificity, thus allowing their use in numerous *in vivo* gene therapy techniques (Hudry & Vandenberghe, 2019; Li & Samulski, 2020). Many treatments targeting neurodegenerative diseases have focused on AAV9 because preclinical trials in various animal models have shown

[1]Genetic Disease Research Branch, National Human Genome Research Institute, National Institutes of Health, Bethesda, MD, USA [2]Division of Biology and Biological Engineering, California Institutes of Technology, Pasadena, CA, USA [3]Department of Medicine, Washington University School of Medicine, St. Louis, MO, USA [4]Medical Genomics and Metabolic Genetics Branch, National Human Genome Research Institute, National Institutes of Health, Bethesda, MD, USA

Correspondence: venditti@mail.nih.gov; bpavan@mail.nih.gov
Alana L Gibson's present address is Division of Biological Sciences, University of California San Diego, San Diego, CA, USA
Tansy Gu's present address is Reproductive and Developmental Biology, National Institute of Environmental Health Sciences, Durham, NC, USA
Benjamin E Deverman's present address is Broad Institute, Boston, MA, USA
Keith Beadle's present address is OHSU Knight Cancer Institute, School of Medicine, Oregon Health and Science University, Portland, OR, USA
Jorge L Rodriguez-Gil's present address is Stanford University School of Medicine, Division of Medical Genetics, Stanford, CA, USA
Daniel S Ory's present address is Casma Therapeutics, Boston, MA, USA
*Cristin D Davidson, Alana L Gibson, and Tansy Gu contributed equally to this work

that it crosses the blood–brain barrier (BBB), efficiently transduces cells in the central nervous system (CNS), and can be safely administered in nonhuman primates (Bevan et al, 2011; Samaranch et al, 2012; Saraiva et al, 2016; Lykken et al, 2018; Hudry & Vandenberghe, 2019). Furthermore, AAV9-derived vectors have already achieved clinical translation, with the successful FDA approval of Zolgensma representing the first use of an AAV vector for the treatment of spinal muscular atrophy type 1, a fatal neurodegenerative disorder of infancy and childhood (Saraiva et al, 2016; Lykken et al, 2018; Al-Zaidy et al, 2019; Lowes et al, 2019). Use of AAV9-based vectors has been extended to clinical trials for other disorders, including giant axonal neuropathy, Pompe disease, GM1 gangliosidosis, Duchenne muscular dystrophy, Batten disease, Danon disease, Gaucher disease, and Mucopolysaccharidosis type IIIA.

Despite theoretical limitations to the application of gene therapy to treat lysosomal storage diseases which feature cell autonomous pathology, AAV9 vectors can be effective in alleviating phenotypes of NPC1 in mouse models. We previously reported the systemic delivery of an AAV9-EF1a(s)-hNPC1 vector significantly increased survival and delayed disease progression in the $Npc1^{m1N/m1N}$ null mouse model (Chandler et al, 2017), even after treatment of the mice as juveniles. Subsequent studies similarly noted improvement in $Npc1^{m1N/m1N}$ mice after intra-cardiac delivery of an AAV9-CMV-NPC1 vector (Xie et al, 2017) or intracerebroventricular delivery of an AAV9-hSynapsin-NPC1 vector delivered in the neonatal period (Hughes et al, 2018). Although encouraging, these results also indicate that careful optimization can significantly improve the therapeutic efficacy of gene therapy in NPC1.

The promising results of the aggregate AAV9 studies have led us to explore AAV9 capsid variants with improved CNS penetration from the systemic circulation as a means to increase the potency of AAV gene therapy for NPC disease. This is particularly important for the more complex NPC disease type 1 as opposed to type 2, where the secretion of soluble NPC2 provides the added benefit of cross-correction (Markmann et al, 2018). Moreover, neuronal deficiency of NPC1 has proven sufficient to mediate CNS disease (Yu et al, 2011), further highlighting the need for vectors with enhanced neuronal tropism. The prototypical capsid, AAV-PHP.B, emerged from an in vivo screen and displayed ~40-fold greater gene transfer efficiency to the central nervous system (CNS) with transduction of a variety of cell types including astrocytes, oligodendrocytes, and neurons after peripheral injection (Deverman et al, 2016). Studies with AAV-PHP.B vectors have demonstrated widespread and efficient gene delivery to the rodent CNS and alleviation of neurological phenotypes (Jackson et al, 2016; Gao et al, 2017; Morabito et al, 2017; Zelikowsky et al, 2018; Lim et al, 2019; Luoni et al, 2020; Yang et al, 2020). However, the ability of systemically administered AAV-PHP.B to transduce the CNS in mice is dependent on the presence of a strain-specific haplotype that includes the gene encoding the GPI-linked protein $Ly6a$; without this permissive $Ly6a$ allele, AAV-PHP.B CNS transduction is severely limited (Hordeaux et al, 2019; Huang et al, 2019; Batista et al, 2020). Importantly, studies with AAV-PHP.B and other engineered capsids (Deverman et al, 2016; Tordo et al, 2018; Wang et al, 2018; Hanlon et al, 2019) can provide preclinical paradigms for future applications of AAV serotypes and/or engineered capsids that display enhanced CNS transduction in other species, including non-human primates (Ivanchenko et al, 2020).

In this study, we compared AAV-PHP.B and AAV9 vectors in the well-established NPC1 null mouse model, $Npc1^{m1N}$. Transgene expression of GFP reporter constructs showed higher transduction throughout the brain for the AAV-PHP.B vector in comparison to AAV9, particularly notable in the hippocampus and midbrain regions. Comparative analyses of $Npc1^{m1N/m1N}$ mice that received comparable doses in otherwise identical vectors showed that mice treated with the AAV-PHP.B-$NPC1$ construct lived longer and showed greater reduction of disease symptoms. Variability in disease phenotypes and CNS copy numbers of AAV-PHP.B–treated mice correlated with the segregation of the permissive and restrictive alleles of $Ly6a$. Interestingly, despite the superior performance of the AAV-PHP.B-$NPC1$ vector in improving survival, weight, and behavior, clear reduction in pathology within the CNS in comparison to the AAV9-$NPC1$ vector was difficult to detect. Both vectors exhibited only moderate correction of brain disease pathology (cholesterol accumulation, inflammation, loss of cerebellar Purkinje cells) compared to untreated $Npc1^{m1N/m1N}$ mice. Overall, results from this proof of concept study suggest that relatively small numbers of CNS cells were effectively transduced by AAV-PHP.B, and this moderate cell correction was enough to markedly improve disease phenotypes, including survival, in $Npc1^{m1N/m1N}$ mice.

# Results

### AAV-PHP.B-GFP reporter construct showed biodistribution throughout the brain in $Npc1^{m1N/m1N}$ mice

Transduction efficacy of AAV-PHP.B versus AAV9 vectors was compared in $Npc1^{m1N/m1N}$ and $Npc1^{+/+}$ mice treated with retro-orbital injections of either an AAV-PHP.B-GFP or AAV9-GFP vector. Each vector contained a GFP reporter construct under control of the ubiquitous elongation factor 1a (shortened) promoter (AAV-PHP.B-EF1a(s)-GFP and AAV9-EF1a(s)-GFP, respectively). Mice were given $1.21 \times 10^{12}$ genome copies (GC) (~$8.55 \times 10^{13}$ GC/kg) of each vector construct between postnatal (P) days 24–27, and GFP expression was analyzed 5.5 wk later (~P63/9 wk). Mice receiving AAV-PHP.B-GFP showed GFP expression throughout the brain (Fig 1A, C, and E), and this expression was greater than that seen in mice given AAV9-GFP (Fig 1B, D, and F). GFP expression was particularly high in the hippocampus, striatum, molecular cell layer of the cerebellum, and vestibular nucleus of $Npc1^{m1N/m1N}$ mice that received the AAV-PHP.B-GFP vector relative to mice that received AAV9-GFP. Similar results were seen in $Npc1^{+/+}$ mice that received AAV-PHP.B-GFP or AAV9-GFP (data not shown). These findings are consistent with previous studies that suggest greater CNS transduction by AAV-PHP.B vectors in comparison to AAV9 vectors, and these results also demonstrate that the underlying disease state does not interfere with AAV-PHP.B CNS transduction.

### $Npc1^{m1N/m1N}$ mice treated with an AAV-PHP.B-$NPC1$ vector showed increased survival and delayed disease phenotype progression

To compare the efficacy of AAV-PHP.B to AAV9, $Npc1^{m1N/m1N}$ mice were treated with identical transgenes in the different capsids. The vectors express $NPC1$ under control of the EF1a (shortened)

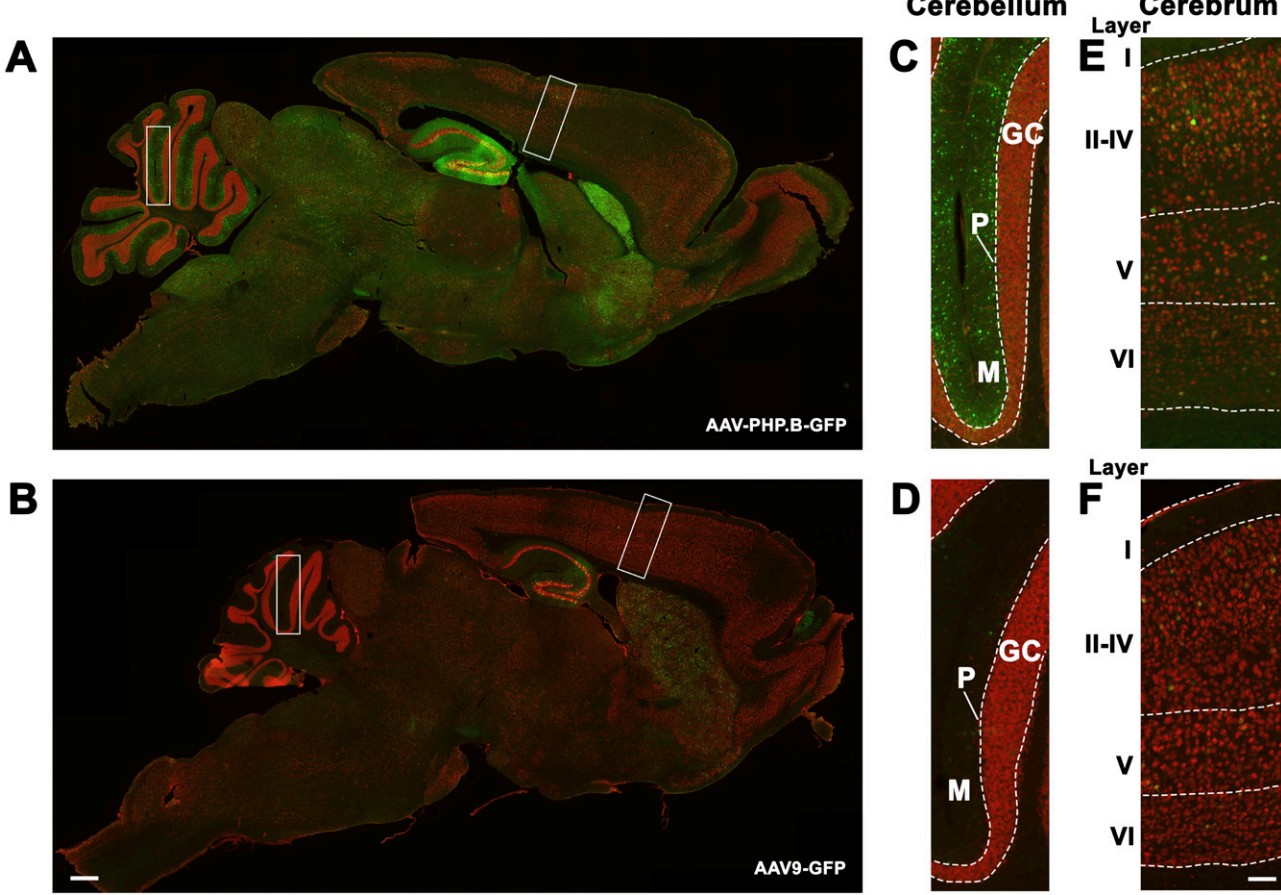

**Figure 1. Biodistribution of GFP expression in *Npc1*[m1N/m1N] mice that received either adeno-associated virus (AAV)-PHP.B-GFP or AAV9-GFP vectors.**
**(A, B)** Sagittal brain sections with immunohistochemical staining of GFP (green) and neuronal nuclei (red). *Npc1*[m1N/m1N] mice treated with AAV-PHP.B-GFP showed increased GFP expression compared to *Npc1*[m1N/m1N] mice treated with AAV9-GFP. **(C, D, E, F)** Higher magnification views of lobule IV/V in the cerebellum and the dorsal neocortex. Vector was administered to mice between 24 and 27 d old and tissues were analyzed at 9 wk. Scale bars = 500 (A, B) and 100 μm (C, D, E, F).

promoter (AAV-PHP.B-EF1a(s)-hNPC1 or AAV9-EF1a(s)-hNPC1, here-after referred to as AAV-PHP.B-*NPC1* or AAV9-*NPC1*, respectively). Three cohorts of *Npc1*[m1N/m1N] mice were injected, as follows: nine mice received AAV-PHP.B-*NPC1* vector at 1.43 × 10$^{12}$ GC (~1.24 × 10$^{14}$ GC/kg), nine mice received AAV9-*NPC1* vector at 1.84 × 10$^{12}$ GC (~1.42 × 10$^{14}$ GC/kg), and five mice received saline-only injections. Technical variability with vector titer assays led to these moderately different doses for AAV-PHP.B-*NPC1* versus AAV9-*NPC1*. Comparison of survival among these three groups (Fig 2A and B) showed that *Npc1*[m1N/m1N] mice treated with AAV-PHP.B-*NPC1* exhibited a signifi-cantly higher median survival of 33.4 wk in comparison to mice treated with AAV9-*NPC1*, for which median survival was only 16 wk (*P* < 0.005, Mantel–Cox log rank test). Both vectors showed signifi-cantly higher median survival than saline-injected *Npc1*[m1N/m1N] mice (*P* < 0.0001), which had all reached terminal end point by 11 wk.

Normal disease progression in *Npc1*[m1N/m1N] mice includes a marked decline in weight starting at about 6 wk of age; therefore, the week at which mice in both vector-treated cohorts reached peak weight was monitored. *Npc1*[m1N/m1N] mice receiving AAV-PHP.B-*NPC1* reached peak weight significantly later than *Npc1*[m1N/m1N] mice receiving AAV9-*NPC1* (mean of 14.3 ± 5.2 versuss 8.1 ± 1.6 wk,

respectively, Fig 2C). In addition, the age of peak weight of AAV-PHP.B-*NPC1*–treated *Npc1*[m1N/m1N] mice was significantly different from that of saline-injected *Npc1*[m1N/m1N] mice, which showed a peak weight at 6.8 ± 0.45 wk (Fig 2C). Longitudinal weight data are shown in Fig S1A and B (males and females, respectively), illus-trating the longer maintenance of weight and longer lifespan of AAV-PHP.B-*NPC1*–treated mice. The lifespan of AAV-PHP.B-*NPC1*–treated mice did not correlate with their weight at the time of injection, suggesting no dosage effect for these mice, although a correlation was seen in AAV9-*NPC1*–treated mice (r = −.7311, *P* = 0.0308, Fig S2). Interestingly, the percent of weight change between 6 and 9 wk of age (Fig 2D) showed that whereas saline-injected *Npc1*[m1N/m1N] mice lost weight, AAV-PHP.B-*NPC1*–treated *Npc1*[m1N/m1N] mice and normal *Npc1*[+/+] mice gained weight at a similar rate during this time period (AAV-PHP.B–treated *Npc1*[m1N/m1N] mean = 12.3% ± 5.6%; *Npc1*[+/+] mean = 12.1% ± 3.0%). In contrast, the cohort of AAV9-*NPC1*–treated mice showed wide variability (mean = 0.88% ± 17.3%, Fig 2D).

To compare the effects of AAV-PHP.B-*NPC1* and AAV9-*NPC1* on *Npc1*[m1N/m1N] disease-associated traits, two behavioral assays were used: phenotype score and balance beam. The phenotype score

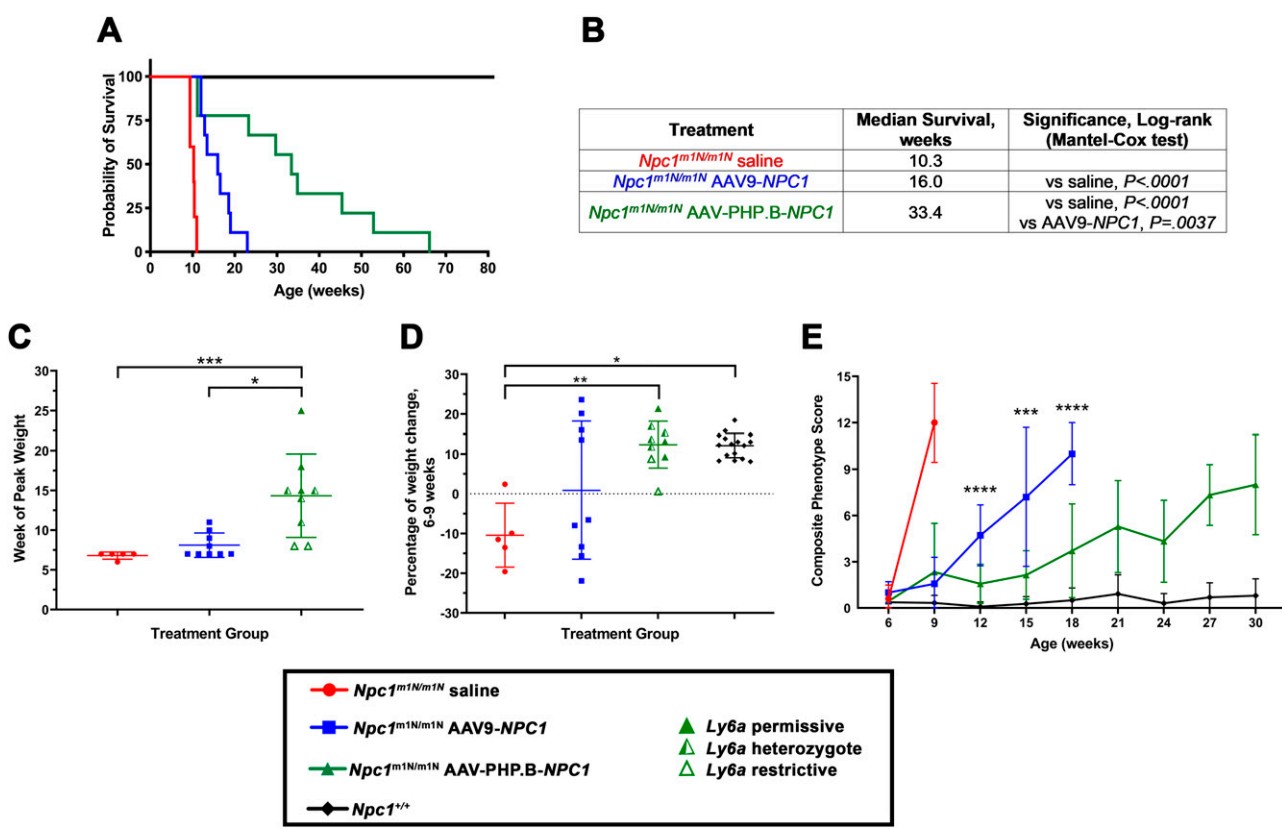

**Figure 2.** *Npc1*[m1N/m1N] mice treated with an *NPC1* adeno-associated virus (AAV)-PHP.B vector showed increased survival and delayed disease phenotype progression.
**(A)** Kaplan–Meier curve depicts survival of the following: *Npc1*[m1N/m1N] mice treated with an AAV-PHP.B vector containing human *NPC1* ($1.43 \times 10^{12}$ genome copy), *Npc1*[m1N/m1N] mice treated with an AAV9 vector containing human *NPC1* ($1.84 \times 10^{12}$ genome copy), saline-injected *Npc1*[m1N/m1N] mice, and untreated *Npc1*[+/+] controls. All mice were administered retro-orbital injections between P24 and P27. **(A, B)** Table of treatment group, median survival, and significance (Mantel–Cox log rank test) of data shown in (A). **(C)** AAV-PHP.B-*NPC1*–treated *Npc1*[m1N/m1N] mice reached peak weight at a significantly older age than saline-injected *Npc1*[m1N/m1N] mice and *Npc1*[m1N/m1N] mice receiving the AAV9-*NPC1* vector (Kruskal–Wallis test with Dunn's multiple comparisons test). AAV9-treated *Npc1*[m1N/m1N] mice showed no significant difference from saline-injected mice. **(D)** Graphical depiction of the percentage of weight change between 6 and 9 wk of age. AAV-PHP.B-*NPC1*–treated *Npc1*[m1N/m1N] mice and normal *Npc1*[+/+] mice gained weight at a similar rate during this time period, both of which were significantly different from the marked weight loss exhibited by saline-injected *Npc1*[m1N/m1N] mice (Welch's ANOVA test with Dunnett's multiple comparisons test). AAV9-*NPC1*–treated mice showed wide variability spanning across the other three groups.
**(E)** Composite phenotype scores for each treatment group, measured at 3-wk intervals starting at 6 wk of age. *Npc1*[m1N/m1N] mice treated with AAV-PHP.B-*NPC1* maintained significantly lower composite scores compared to *Npc1*[m1N/m1N] mice treated with AAV9-*NPC1* from weeks 12–18 (two-way ANOVA with Tukey's multiple comparisons test). A full table of 2-way ANOVA results is presented in Table S1. Composite phenotype scores include disease-relevant measures of gait, kyphosis, ledge test, hind limb clasp, grooming, and tremor, with a higher score indicating a more severe disease phenotype. **(A, B, C, D)** For panels (A, B, C, D), n's are as follows: *Npc1*[m1N/m1N] saline = 5, *Npc1*[m1N/m1N] AAV9-*NPC1* = 9, *Npc1*[m1N/m1N] AAV-PHP.B-*NPC1* = 9, *Npc1*[+/+] = 14. **(E)** Of note for panel (E), the n of mice at each time point in all three *Npc1*[m1N/m1N] groups became smaller at later time periods because of animals reaching end stage (see Table S1). *$P < 0.05$, **$P < 0.01$, ***$P < 0.001$, ****$P < 0.0001$.

assesses six measures of the NPC disease phenotype (gait, kyphosis, ledge test, hind limb clasp, grooming, and tremor) while the balance beam is an indicator of motor coordination. A higher composite score for the phenotype assay or greater number of slips for the balance beam correlates with a worsened disease state. From weeks 12–18, *Npc1*[m1N/m1N] mice treated with AAV-PHP.B-*NPC1* maintained significantly lower composite phenotype scores than *Npc1*[m1N/m1N] mice treated with AAV9-*NPC1* ($P < 0.001$, two-way ANOVA with Tukey's multiple comparisons test, Fig 2E). Two additional comparisons showed treatment with AAV-PHP.B-*NPC1* delayed disease progression: AAV-PHP.B-*NPC1*–treated mice had a composite score that did not differ from *Npc1*[+/+] control mice at weeks 12 and 15, and from weeks 18–24, AAV-PHP.B-*NPC1*–treated mice maintained scores that were better than those of end-stage AAV9-*NPC1*–treated mice (Table S1). Similarly, AAV-PHP.B-*NPC1*–treated mice showed delayed loss of motor coordination on the balance beam assay (Fig S3). From weeks 12–18, *Npc1*[m1N/m1N] mice treated with AAV-PHP.B-*NPC1* had fewer slips on the balance beam than *Npc1*[m1N/m1N] mice treated with AAV9-*NPC1* ($P < 0.01$, two-way ANOVA with Tukey's multiple comparisons test; Table S1). In addition, enhanced therapeutic benefit of AAV-PHP.B-*NPC1* on ambulation and ataxia relative to AAV9-*NPC1* can be appreciated, as demonstrated in Videos 1 and 2.

## Differential impact of AAV-PHP.B and AAV9 vectors on *NPC1* transduction efficiency in brain and liver

Further analyses were performed on *Npc1*[m1N/m1N] mice to examine the transduction efficiency of AAV-PHP.B-*NPC1* and AAV9-*NPC1* vectors. Transduction efficiency of each vector was evaluated by measuring *NPC1* copy number in cerebrum, cerebellum, and liver by droplet digital PCR (ddPCR) at 9 wk of age and at end stage. At 9 wk,

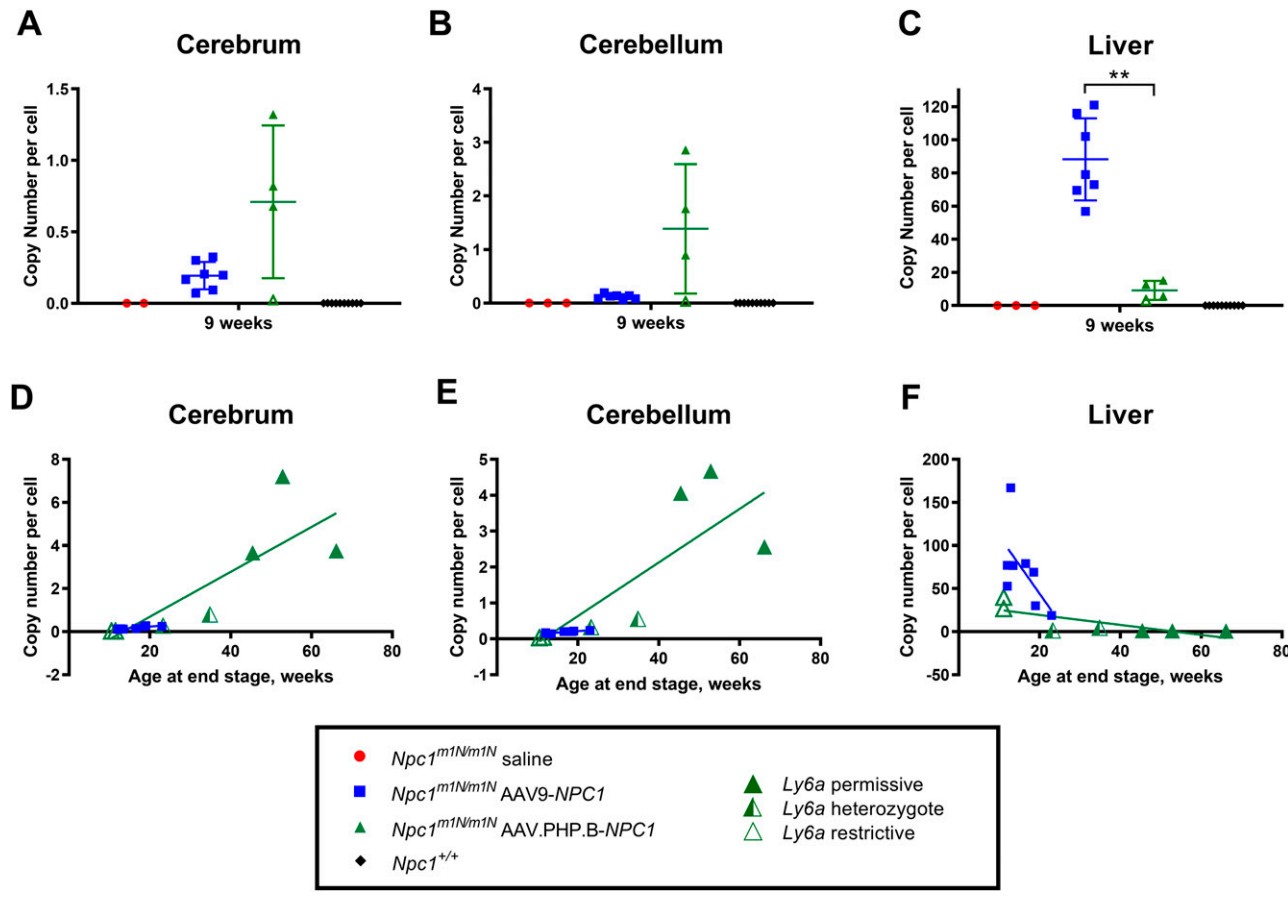

**Figure 3. Differential transduction efficiency of adeno-associated virus (AAV)-PHP.B-*NPC1* and AAV9-*NPC1* vectors in *Npc1^{m1N/m1N}* mice in brain and liver.**
**(A, B, C)** Droplet digital PCR was used to measure *NPC1* copy number at 9 wk of age. There was a trend for higher copy number in cerebrum and cerebellum of AAV-PHP.B-*NPC1*- compared with AAV9-*NPC1*–treated *Npc1^{m1N/m1N}* mice. In contrast, liver tissue showed the opposite result, with significantly lower copy numbers in *Npc1^{m1N/m1N}* mice treated with AAV.PHP.B-*NPC1* (unpaired *t* test). **(D, E, F)** *NPC1* copy number of AAV9-*NPC1*– and AAV-PHP.B-*NPC1*–treated *Npc1^{m1N/m1N}* mice graphed as a function of age at the end stage. The AAV-PHP.B-*NPC1*–treated mice with the highest *NPC1* copy number in cerebrum and cerebellum also survived the longest. *Ly6a* allelic determination is shown for AAV-PHP.B-*NPC1*–treated *Npc1^{m1N/m1N}* mice as well. Both groups showed a positive correlation between age at end stage and *NPC1* copy numbers in cerebrum (AAV-PHP.B-*NPC1*, r = 0.929, *P* = 0.007; AAV9-*NPC1*, r = 0.886, *P* = 0.006) and cerebellum (AAV-PHP.B-*NPC1*, r = 0.893, *P* = 0.012; AAV9-*NPC1*, r = 0.778, *P* = 0.03; Spearman's correlation coefficient test). AAV-PHP.B-*NPC1* copy number in liver did not positively correlate with age at the end stage.

AAV-PHP.B-*NPC1*–treated mice showed higher *NPC1* copy numbers than AAV9-*NPC1*-transduced mice in both brain regions; however, these differences were not statistically significant likely because of small sample size and variability (Fig 3A and B). In contrast, liver showed significantly higher *NPC1* copy number in *Npc1^{m1N/m1N}* mice that received AAV9-*NPC1* in comparison to AAV-PHP.B-*NPC1* (Fig 3C and *P* < 0.001, unpaired *t* test). These patterns of relatively higher AAV-PHP.B-*NPC1* brain transduction and relatively higher AAV9-*NPC1* liver transduction persisted in end stage *Npc1^{m1N/m1N}* mice (Fig S4). Interestingly, positive correlations were seen between age at end stage disease and *NPC1* copy numbers in cerebrum and cerebellum and in both the AAV-PHP.B-*NPC1*– and AAV9-*NPC1*–transduced mice (Fig 3D and E), suggesting the levels of *NPC1* transduction in the brain were important for extending lifespan. In contrast, the liver *NPC1* copy number did not positively correspond to lifespan in either group of mice (Fig 3F).

The higher transduction of AAV-PHP.B-*NPC1* in brain was corroborated by assessment of NPC1 protein levels. Western blot analysis of cerebrum tissue collected from AAV-PHP.B-*NPC1*–

transduced *Npc1^{m1N/m1N}* mice showed modest levels of NPC1 protein with the highest *NPC1* copy number and longest survival (Fig 4A), whereas immunoreactive NPC1 was undetectable in mice with the lowest vector copy number and shortest lifespan (data not shown). None of the AAV9-*NPC1*–transduced *Npc1^{m1N/m1N}* mice showed detectable NPC1 by Western blot (Fig 4A). In contrast, Western blot analyses of liver revealed that AAV9-*NPC1*–transduced *Npc1^{m1N/m1N}* mice showed consistently higher protein levels of NPC1 than AAV-PHP.B-*NPC1*–transduced mice (Fig 4B). Similar to cerebrum, liver showed a correspondence between the *NPC1* copy numbers found in ddPCR analysis and the presence or absence of NPC1 protein on Western blots.

### Allelic differences at the Ly6a locus are associated with the variable phenotypes of AAV-PHP.B-*NPC1*–treated mice

Notable variability was present in lifespan, phenotype score, and copy number in brain tissues of AAV-PHP.B-*NPC1*–treated *Npc1^{m1N/m1N}* mice (Figs 2 and 3). After the completion of the AAV-PHP.B-*NPC1*

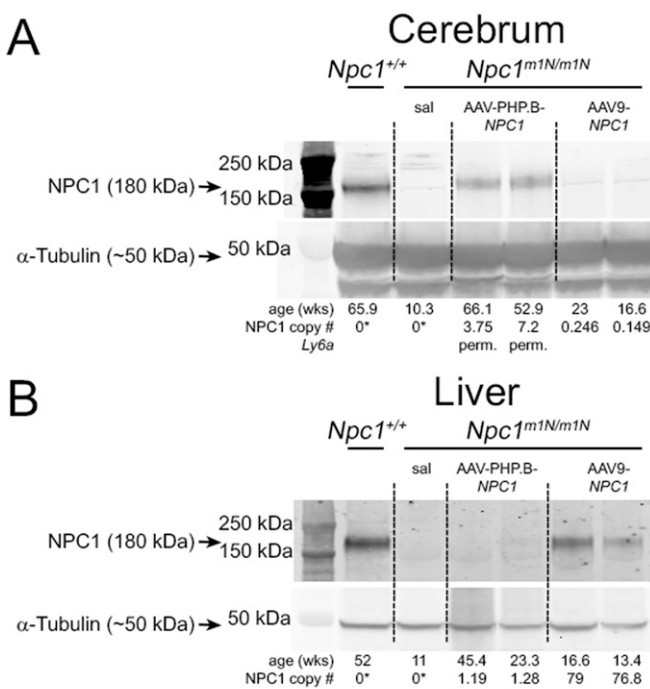

**Figure 4. NPC1 protein levels correspond with differential transduction efficiency of adeno-associated virus (AAV)-PHP.B-*NPC1* and AAV9-*NPC1* vectors.**

**(A, B)** Western blots were used to confirm the presence or absence of NPC1 protein in cerebrum and liver tissue. The *Npc1^m1N/m1N^* model is a null, thus the only NPC1 protein present would arise from the transduced vectors. Age in weeks and *NPC1* copy number for each mouse is under α-tubulin loading control. **(A)** In cerebrum, NPC1 protein was detectable only in the longest surviving AAV-PHP.B-*NPC1*–treated *Npc1^m1N/m1N^* and *Npc1^+/+^* mice, and was not detected in AAV9-NPC1–treated *Npc1^m1N/m1N^* mice. **(B)** Conversely, analysis of liver showed that NPC1 protein was present in most AAV9-*NPC1*–treated *Npc1^m1N/m1N^* mice, but only rarely in *Npc1^m1N/m1N^* mice receiving AAV-PHP.B-*NPC1*. *Droplet digital PCR assayed only human *NPC1*, not murine NPC1, hence zero values for non-gene therapy treated mice. Full unedited gels for Fig 4: Please see green outlines below for cerebrum and blue outlines below for liver to denote the portion of the gel used in Fig 4. Each gel was co-labeled with NPC1 and α-tubulin and different secondaries were used for each primary antibody.
Source data are available for this figure.

experiments, other groups published work identifying strain-specific effects on the CNS transduction efficiency of AAV-PHP.B that are associated with different haplotypes at *Ly6a*, a gene encoding a GPI-anchored protein expressed at the BBB (Hordeaux et al, 2019; Huang et al, 2019; Batista et al, 2020). These *Ly6a* studies showed that BALB/cJ mice had lower AAV-PHP.B transduction efficiency, of relevance because the spontaneous *Npc1^m1N^* allele arose on a BALB/c-derived strain (Morris et al., 1977, 1982). Therefore, we sought to examine the *Ly6a* genotype in the *Npc1^m1N/m1N^* mice used in our experiments.

Genotype analysis was performed for two exonic SNPs in *Ly6a* that exhibit nonsynonymous variants (relative to C57Bl6/J reference sequence) in BALB/cJ and other inbred strains, as follows: rs32279213, p.D63G, and rs213983347, p.V106A. Two haplotypes that include these SNPs associate with brain transduction levels of AAV-PHP.B that are either higher (permissive genotype, p.D63 and p.V106) or lower (restrictive genotype, p.G63 and p.A106) (Hordeaux

et al, 2019; Huang et al, 2019; Batista et al, 2020). Interestingly, genotyping results revealed that both *Ly6a* genotypes were segregating in AAV-PHP.B–treated *Npc1^m1N/m1N^* mice (Table 1), and the *Ly6a* genotypes correlated with the biodistribution of GFP, lifespan and *NPC1* copy number in the brain. Enhanced biodistribution of the GFP vector (data not shown), as well as increased lifespan and greater *NPC1* copy number occurred in the AAV-PHP.B-*NPC1*–treated *Npc1^m1N/m1N^* mice that were homozygous for the permissive *Ly6a* genotype (Table 1 and Fig 3D–F). In contrast, AAV-PHP.B-*NPC1*–treated *Npc1^m1N/m1N^* mice homozygous for the restrictive *Ly6a* genotype showed the shortest lifespan and lowest *NPC1* copy number (Table 1). These findings suggest the variability of phenotype improvement in AAV-PHP.B–treated mice was associated with their *Ly6a* genotype. A genome-wide scan of two randomly selected *Npc1^m1N/m1N^* mice in this study also showed the presence of BALB/cJ and non-BALB/cJ genetic markers (homozygous BALB/cJ markers were present for only 76%–81% of the informative markers analyzed). This genetic variability could be specific to our colony, as genotyping from the Jackson Laboratory colony (Stock #003092, BALB/cNctr-Npc1<m1N>/J) showed these mice were homozygous for the restrictive *Ly6a* genotype. Moreover, two distinct cohorts from collaborative studies done at the Jackson Laboratory in the *Npc1^m1N/m1N^* mice also revealed only the restrictive *Ly6a* genotype (Pavan, unpublished observation).

## Differential impact of NPC1 AAV-PHP.B-*NPC1* and AAV9-*NPC1* vectors and Ly6a genotype on *Npc1^m1N/m1N^* metabolomics

A hallmark of disease progression in NPC1 model mice is the accumulation of complex sphingolipid species and cholesterol oxidation products in various tissues as disease severity progresses (Pentchev et al, 1980; Goldin et al, 1992; Fan et al, 2013; Praggastis et al, 2015). Therefore, a broad panel of metabolites derived from cholesterol and sphingolipids was examined in cerebrum, cerebellum, and liver tissue from end stage AAV9-*NPC1*– and AAV-PHP.B-*NPC1*–treated *Npc1^m1N/m1N^* mice and compared with *Npc1^+/+^* and *Npc1^m1N/m1N^* saline-injected mice (Fig S5). No differences were apparent between the saline-injected and AAV9-*NPC1*–treated *Npc1^m1N/m1N^* mice. The shortest-lived AAV-PHP.B-*NPC1*–treated *Npc1^m1N/m1N^* mice also did not show differences from untreated *Npc1^m1N/m1N^* mice. No consistent changes in cholesterol levels were apparent in the brain or liver tissues of AAV9-*NPC1*– and AAV-PHP.B-*NPC1*–treated *Npc1^m1N/m1N^* mice. Interestingly, the three longest lived AAV-PHP.B-*NPC1*–treated *Npc1^m1N/m1N^* mice with a homozygous permissive *Ly6a* genotype showed lower levels of sphingosine, sphinganine, and 3-keto-sphinganine in both cerebellum and cerebrum, suggesting modest correction in the brain (Fig S5A and B). In addition, these three mice showed lower levels than untreated *Npc1^m1N/m1N^* mice of GA2 species and 3β, 5α, 6β-trihydroxycholestane in the cerebellum. Both markers have shown improvement in mice, cats, and patients treated with 2-hydroxypropyl-β-cyclodextrin (HPβCD), a compound evaluated in clinical trial for NPC disease (Tortelli et al, 2014; Ory et al, 2017). In contrast, liver did not show notable correction of any individual lipid species in either AAV9-*NPC1*– or AAV-PHP.B-*NPC1*–treated *Npc1^m1N/m1N^* mice, and the longest lived AAV-PHP.B-*NPC1*–treated *Npc1^m1N/m1N^* mice showed higher levels of most lipids analyzed (Fig S5C). Of note, hierarchical clustering of the

**Table 1.** *Npc1[m1N/m1N]* mice treated with adeno-associated virus-PHP.B show allelic variance at the *Ly6a* locus that correlates with lifespan and copy number per cell.

| Age, wk[a] | SNP genotyping at *Ly6a* locus | | Copy number per cell | | |
|---|---|---|---|---|---|
| | rs32279213[b] | rs213983347[c] | Cerebrum | Cerebellum | Liver |
| 9 | G/G | C/C | 0.0229 | 0.032 | 3.1 |
| 9 | A/G | T/C | 0.678 | 0.899 | 12.7 |
| 9 | A/G | T/C | 1.32 | 1.76 | 5.68 |
| 9 | A/G | T/C | 0.819 | 2.86 | 15.2 |
| 11.1 | G/G | C/C | 0.027 | 0.047 | 39.8 |
| 11.1 | G/G | C/C | 0.0286 | 0.03 | 26.8 |
| 34.9 | A/G | T/C | 0.799 | 0.558 | 4.55 |
| 23.3 | A/G | T/C | 0.278 | 0.329 | 1.28 |
| 33.4 | A/G | T/C | ND | ND | ND |
| 29.7 | A/G | T/C | ND | ND | ND |
| 45.4 | A/A | T/T | 3.66 | 4.06 | 1.19 |
| 66.1 | A/A | T/T | 3.75 | 2.56 | 0.82 |
| 52.9 | A/A | T/T | 7.2 | 4.67 | 1 |

[a]9-wk-old mice were deliberately euthanized at this age; all other ages reflect end stage of disease.
[b]G is the restrictive allele at rs32279213 (encoding p.G63) and A is the permissive allele (encoding p.D63).
[c]C is the restrictive allele at rs213983347 (encoding p.A106) and T is the permissive allele (encoding p.V106).
ND, not determined.

metabolomics data showed that AAV-PHP.B-*NPC1*–treated mice homozygous for the permissive *Ly6a* allele clustered together in cerebrum, cerebellum, and liver (data not shown), further suggesting that there are *Ly6a*-correlated differences in the lipidomic profiles of AAV-PHP.B-*NPC1*–treated *Npc1[m1N/m1N]* mice at end stage.

### Differential impact of *NPC1* AAV-PHP.B and AAV9 vectors on pathology

The impact of gene therapy on accumulation of unesterified cholesterol in the CNS and liver, a pathological hallmark of NPC disease, was evaluated with filipin complex (a macrolide that labels unesterified cholesterol in immunofluorescent staining) (Vanier & Latour, 2015). Additional known pathological changes, such as neuroinflammation and GM2 ganglioside accumulation were also investigated. Hematoxylin & eosin (H&E) staining provided further information with respect to vacuolization of neurons in the hippocampus and Kupffer cells (KCs) in the liver.

Filipin staining revealed prominent accumulation of unesterified cholesterol in the cerebellum in saline-injected *Npc1[m1N/m1N]* mice when compared to *Npc1[+/+]* mice, as highlighted in Lobules III/IV (Fig 5A and B, *Npc[+/+]* versus *Npc1[m1N/m1N]*, respectively). After treatment with either AAV9-*NPC1* or AAV-PHP.B-*NPC1*, the anticipated reduction in cholesterol accumulation was modest (Fig 5C–F), a somewhat unexpected result given the significant impact on disease course. Even in regions with high expression observed in GFP studies, such as the hippocampus, reduction of pathology was minimal (Fig S6A–H). Only modest improvement of gliosis in gene therapy treated mice compared with the saline-injected *Npc1[m1N/m1N]* cohort was observed, as evidenced by microglial and astrocytic staining (anti-IBA1 and anti-GFAP, respectively). Quantification of

microgliosis, or the percentage of IBA1[+] area, was determined in lobules III, VI/VII, and IX of cerebellar tissue sections. Saline-injected *Npc1[m1N/m1N]* mice displayed a significantly higher percentage IBA1[+] area than *Npc1[+/+]* mice (Fig S7A; Kruskal–Wallis test with Dunn's multiple comparison post-test, *P* = 0.0029 for lobule III, *P* = 0.0034 for lobules VI/VII, *P* = 0.0264 for lobule IX). IBA1[+] area in specified lobules of cerebella from *Npc1[m1N/m1N]* mice treated with either gene therapy vector were not significantly different from saline-injected *Npc1[m1N/m1N]* or *Npc1[+/+]* mice, although AAV9-PHP.B-*NPC1*–treated *Npc1[m1N/m1N]* mice did trend toward greater reduction in pathology. This finding correlates with the higher AAV9-PHP.B-*NPC1* vector copy number and NPC1 protein levels in brain. Overall, when considering the numerous different cell types in the brain, NPC1-associated pathology minimally improved after gene therapy treatment.

Filipin and immunofluorescent staining with anti-CD68 was performed on 9-wk-old liver tissue to visualize unesterified cholesterol accumulation and KCs, respectively (Fig 6). Untreated *Npc1[+/+]* mice showed no cholesterol accumulation and normal cellular architecture of hepatocytes and KCs (Fig 6A, E, and I), whereas age-matched, saline-injected *Npc1[m1N/m1N]* mice showed extensive cholesterol accumulation and vacuolization in both hepatocytes and KCs (Fig 6B, F, and J). 9-wk-old *Npc1[m1N/m1N]* mice treated with AAV9-*NPC1* and AAV-PHP.B-*NPC1* showed similar relative abundance of lipid-laden KCs (green, Fig 6G and H) that corresponded to the very bright, rounded filipin positive cells (Fig 6C and D, depicting filipin channel alone). The only noticeable, although modest, pathological improvement in liver was found in AAV9-*NPC1*–treated *Npc1[m1N/m1N]* mice. Livers from these mice presented clusters of hepatocytes, frequently found near portal veins, that were free from cholesterol accumulation and vacuolization (Fig 6C, arrows

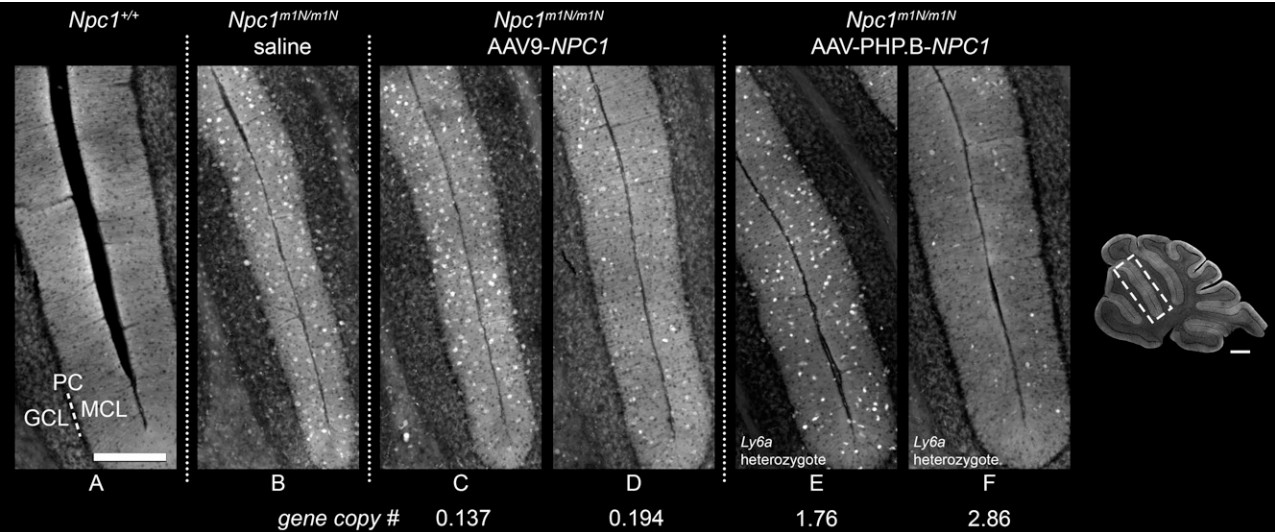

**Figure 5. Impact of adeno-associated virus (AAV)9-*NPC1* and AAV-PHP.B-*NPC1* vectors on cerebellar pathology.**
**(A, B, C, D, E, F)** Cholesterol accumulation (visualized by Filipin labeling) is seen as white punctae, predominantly in cells of the molecular layer, of lobules III/IV of the cerebellum in saline and gene therapy–treated *Npc1^{m1N/m1N}* mice **(B, C, D, E, F)**. Note heterogeneity of modest correction observed in both AAV9-*NPC1*– and AAV-PHP.B-*NPC1*–transduced mice **(C, D, E, F)**. Abbreviations in panel (A): GCL, granule cell layer; PC, Purkinje cell layer (dotted line); MCL, molecular cell layer. Scale bars = 500 µm (inset of cerebellum) and 250 µm (A, B, C, D, E, F).

and Fig 6K). On the other hand, livers from AAV-PHP.B-*NPC1*–treated *Npc1^{m1N/m1N}* mice (Fig 6D, H, and L) showed very little correction of the storage phenotype, with only the occasional filipin-negative hepatocyte, consistent with the much lower copy number seen with ddPCR. Comparison of H&E staining for several mice from each treatment and age-group revealed heterogeneous pathological changes in AAV-PHP.B-*NPC1*–treated *Npc1^{m1N/m1N}* mice, attributable to the *Ly6a* genotype, and extensive vacuolization in both gene therapy treatment groups at end stage disease (Fig S8; A-P 9-wk-old time point, Q-X end stage time point; column 1–4: *Npc1^{+/+}* mice, *Npc1^{m1N/m1N}* saline-injected mice, *Npc1^{m1N/m1N}* AAV9-*NPC1*-injected mice, and *Npc1^{m1N/m1N}* AAV-PHP.B-*NPC1*-injected mice, respectively). Finally, the percentage of CD68^+ area in sectioned liver tissue was compared between the different treatment groups (Fig S7B). Livers from saline-injected *Npc1^{m1Nm1N}* mice displayed significantly larger CD68^+ areas than did livers from *Npc1^{+/+}* mice (Kruskal–Wallis test with Dunn's multiple comparison post-test; P = 0.0109). Although neither AAV9-*NPC1*- nor AAV-PHP.B-*NPC1*–treated *Npc1^{m1N/m1N}* mice showed significant differences from values for *Npc1^{m1N/m1N}* mice administered saline, there was a trend toward reduced CD68^+ area in the AAV9-*NPC1*–treated mice which correlates with the assessments of vector copy number and NPC1 protein levels in liver.

Whereas toxicity is possible with high-dosage AAV9 (Hinderer et al, 2018; Hordeaux et al, 2018), we found no histological indicators consistent with hepatic injury or genotoxicity. Upon examination of H&E staining for liver in *Npc1^{m1N/m1N}* mice treated with AAV9-*NPC1*, the cytoarchitecture actually looked very similar to *Npc1^{+/+}* liver and showed no obvious signs of toxicity (Fig S8A, E, I, and M versus Fig S8C, G, K, and O; *Npc1^{+/+}* vs. *Npc1^{m1N/m1N}*, respectively). The abnormal cells observed in the AAV9-treated livers are cholesterol-laden KCs as depicted in Fig 6, a known pathology of NPC1 disease.

Furthermore, the mice had no adverse clinical symptoms as have been described in nonhuman primates or piglets treated with high doses of AAV9 early in life (Hinderer et al, 2018).

## Discussion

This study demonstrates that treatment of *Npc1^{m1N/m1N}* mice with an AAV-PHP.B vector containing human *NPC1* significantly increased survival, delayed weight loss, and slowed disease progression compared to mice receiving an AAV9-pseudoserotyped *NPC1* vector. Consistent with previous observations (Deverman et al, 2016), reporter studies revealed that AAV-PHP.B-EF1a(s)-GFP transgene expression was widespread throughout the brain after systemic delivery in both *Npc1^{m1N/m1N}* mice and wild type littermates. The greater transduction efficiency observed with a neurotrophic *NPC1* vector enabled increased CNS correction, and led to prolonged disease amelioration in the *Npc1^{m1N/m1N}* mice. Although the importance of liver in NPC1 disease cannot be minimized, our data also suggest that significant improvement in disease course can be achieved in the absence of substantial liver correction. Finally, although preclinical data from animal models can provide essential proof-of-concept data, it is important to consider the limitations of these model systems, such as species effects of capsids and genetic admixture in mouse models, before translating therapies to the NPC1 patient population.

We noted a superior efficacy of an AAV-PHP.B vector compared with AAV9 in the *Npc1^{m1N/m1N}* mice studied in our colony. This was evident by a significant increase in survival in the treated mutants, an important surrogate for NPC1 disease progression. The longest surviving AAV-PHP.B-*NPC1*–treated *Npc1^{m1N/m1N}* mice were >1 yr old

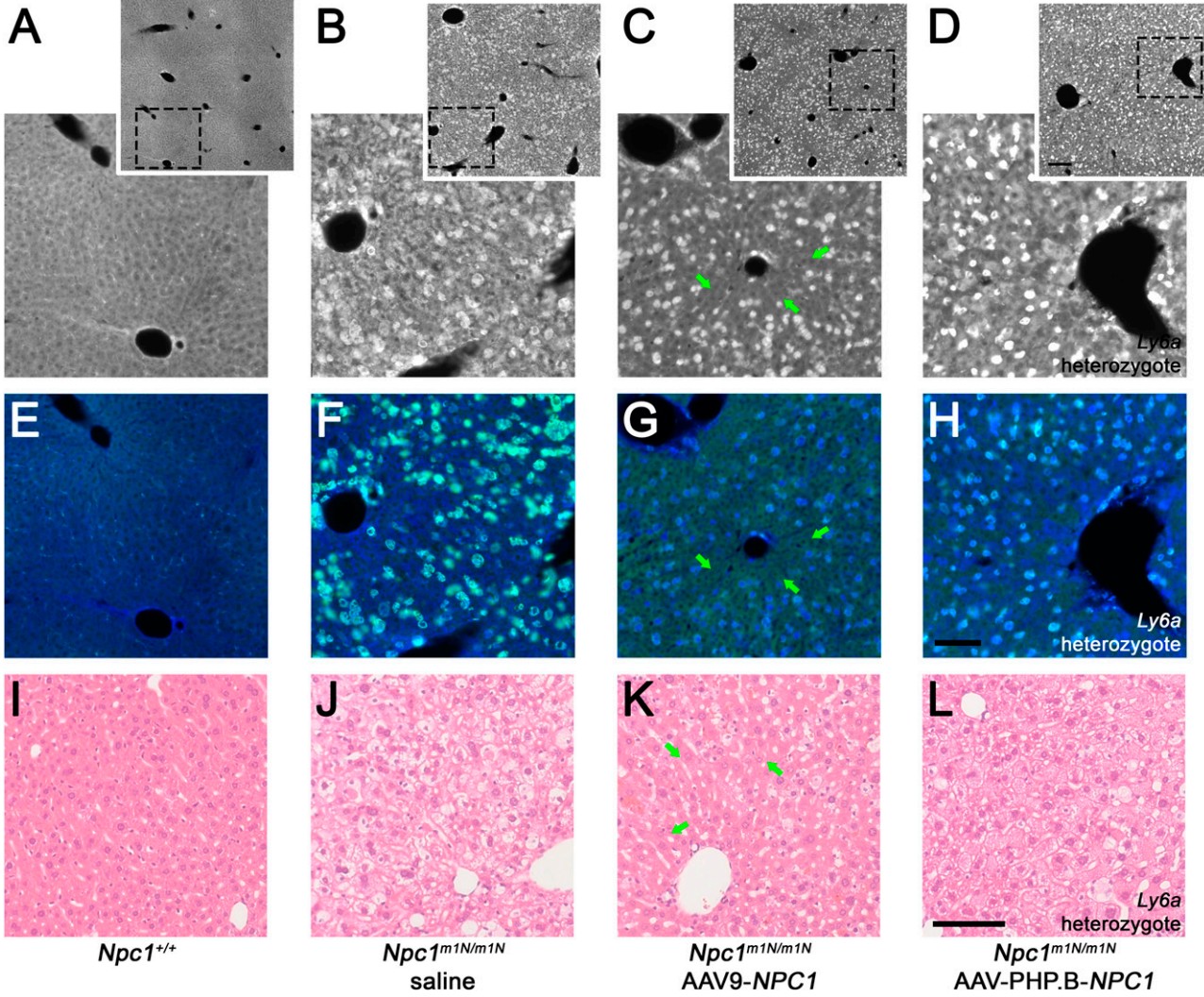

**Figure 6. Differential impact of adeno-associated virus (AAV)9-*NPC1* and AAV-PHP.B-*NPC1* vectors on liver pathology.**
**(A, B, C, D, E, F, G, H)** Cholesterol accumulation (visualized by Filipin labeling: white in top row, blue in middle row) is pronounced in both hepatocytes and Kupffer cells (CD68⁺ green in middle row) of saline and AAV-PHP.B-*NPC1*–treated *Npc1^m1N/m1N* mice (B, D, respectively). **(A, B, C, D)** Insets (A, B, C, D) provide an overview of pathology. **(C, G, K)** Groups of corrected hepatocytes, though not Kupffer cells, are visible in the AAV9-*NPC1*–treated *Npc1^m1N/m1N* mice (arrows in C, G, K) consistent with moderate pathology reduction. **(I, J, K, L)** Hematoxylin and eosin staining support the mildly reduced pathology found in AAV9-*NPC1*–treated *Npc1^m1N/m1N* mice (K) as compared to saline or AAV-PHP.B-*NPC1* treatments (J, L). **(A, E, I)** Normal *Npc1^+/+* liver is shown in (A, E, I) for comparison. Scale bars = 200 μm (insets only, top row) or 100 μm (all other panels).

which, to our knowledge, represents the longest reported lifespan for mutant mice homozygous for this severe *Npc1* allele. In the murine model, a single injection of AAV at weaning yielded survival benefits comparable to 2-hydroxypropyl-β-cyclodextrin, a promising NPC therapeutic which has advanced through a Phase 3 clinical trial (Ory et al, 2017). This cyclodextrin is a small cyclic sugar molecule that requires invasive delivery via the intrathecal route, lifelong dosing, and has significant ototoxicity as a frequent side effect (Ory et al, 2017). Therefore, a single administration of gene therapy, if it had long lasting effects, could represent an important new therapy that might be more effective, and perhaps synergize with other treatments. Combination AAV and cyclodextrin studies are underway, and might help define a new regimen to treat patients, one that hopefully would offer considerable improvement over current investigational or off-label treatments.

The gene therapy studies presented here are consistent with previous conditional and transgenic animal experiments that suggest greater CNS correction can lead to enhanced disease amelioration (Ko et al, 2005; Elrick et al, 2010; Yu et al, 2011). First, gene therapy–treated *Npc1^m1N/m1N* mice with the longest survival had the highest *NPC1* copy number in cerebrum and cerebellum, indicative of greater CNS transduction. Slower deterioration of motor coordination and disease phenotype, as demonstrated by the balance beam and phenotypic screening behavioral assays, was also noted in the mice with higher vector GC numbers. Importantly, AAV-PHP.B-*NPC1*–treated *Npc1^m1N/m1N* mice displayed an even slower progression of disease than did AAV9-*NPC1*–treated *Npc1^m1N/m1N* mice. Finally, maintenance of greater body weight was most apparent in the AAV-PHP.B-*NPC1*–treated *Npc1^m1N/m1N*

mice. All aforementioned improvements were exhibited in the treatment group displaying the highest *NPC1* copy number in brain: AAV-PHP.B-*NPC1*. Although the results were variable, they consistently supported the observation that higher vector copy numbers in the brain correlated with greater improvement.

The extraneuronal disease of NPC1 is clinically significant, with hepatosplenomegaly and persistent liver disease, and even liver failure, documented in NPC1 patients (Kelly et al, 1993; Vanier, 2010; Patterson et al, 2013; Geberhiwot et al, 2018). However, whereas *Npc1*$^{m1N/m1N}$ mice treated with AAV9-*NPC1* showed mild improvement in liver pathology, all other phenotypic measures were reduced compared with the more CNS-trophic AAV-PHP.B vector, highlighting the fact that correction of the CNS, broadly, at a very low level, drives phenotypic correction. Thus, our gene therapy experiments may serve to provide an alternative estimate to the mouse chimera mixing studies, which documented the need for ~30% wild-type cells, to achieve phenotypic correction in Npc1 mice (Ko et al, 2005). Although we are uncertain as to the exact percent and cell type of CNS transduction achieved herein, the reported studies (Fig 1) and Western blotting (Fig 4) suggest it is <30%, and could perhaps inform the selection of a serotype, dose, and route of delivery for future human translation.

The influence of allelic variance at the *Ly6a* locus on treatment efficacy of the AAV-PHP.B vector is well exemplified in our study cohort. The protein encoded by *Ly6a* is expressed at the BBB, and transduction efficiency of AAV-PHP.B correlates with two different haplotypes across the *Ly6a* locus, which are present in various inbred strains of mice. These haplotypes include coding SNPs that may affect *Ly6a* function as well as upstream SNPs that may affect Ly6a protein expression at the BBB (Hordeaux et al, 2019; Huang et al, 2019; Batista et al, 2020). The *Npc1*$^{m1N/m1N}$ colony used in our studies harbors both permissive and restrictive genotypes that correlate with the effects seen in the outcome measures presented. In comparison, limited sampling from the Jackson Laboratory colony revealed only the restrictive genotype (Pavan, unpublished observation). Differences noted in published studies between *Npc1*$^{m1N/m1N}$ mice in distinct facilities, particularly in terms of average survival without therapeutic intervention, might be caused by genetic admixture and/or fixation of modifier alleles, and highlights the importance of exploring strain effects in murine models that are used to generate preclinical enabling data. Recent studies showed reduced efficacy of AAV-PHP.B in nonhuman primates that is likely attributable to the absence of *LY6A* in primates (Hordeaux et al, 2018; Matsuzaki et al, 2018; Liguore et al, 2019), suggesting that novel viral variants may not be readily transferred between species but instead would need to be generated in a species-specific manner. As such, translation from model organism to human must be considered and investigated. For many gene therapy studies, especially those using novel engineered capsids, nonhuman primate studies and relevant human culture models may be needed to validate and optimize a gene therapy vector for delivery to patients with NPC disease and related disorders. Advances in the identification of novel serotypes that cross the BBB in humans, and capsid engineering to derive CNS trophic variants (Castle et al, 2016; Hudry et al, 2018; Sullivan et al, 2018; Hanlon et al, 2019; Havlik et al, 2020) should help improve vectors, as recent studies highlight (Gray et al, 2013; Frederick et al, 2020; Yoon et al, 2020).

In summary, our studies confirm that achieving even moderate transduction of the CNS using an AAV9-*NPC1* vector can have profound effects on disease course, but that much greater correction can be demonstrated with a neurotrophic AAV-PHP.B vector, suggesting that eventual clinical translation may be best accomplished using a capsid that has similar properties in humans.

# Materials and Methods

## Vector construction and production

The transgene EF1a(s)-*NPC1* was previously described (Chandler et al, 2017) and the analogous GFP reporter was prepared by replacing *NPC1* with eGFP to make EF1a(s)-GFP. All therapeutic and control vectors were produced by the Beckman Institute CLOVER Center under direction of Dr V Gradinaru in the Division of Biology and Biological Engineering at the California Institutes of Technology as previously described (Deverman et al, 2016), and serotyped as AAV9 or AAV-PHP.B.

## Animals

Animal work in these studies was performed according to the animal care and use protocols approved by the National Institutes of Health. Heterozygous mice from the BALB/cNctr-*Npc1*$^{m1N}$/J strain were crossed to generate homozygous *Npc1*$^{m1N/m1N}$ mutants and *Npc1*$^{+/+}$ control littermates. Mice were weighed once per week and then more frequently as disease progressed. Euthanasia was performed when end stage disease progression was reached, as determined by the presence of at least two of the following signs: 30% loss of maximum weight, reluctance to move about cage, repeated falling to side during forward ambulation, and palpebral closure/eyes appearing dull rather than bright.

## Study design (based on guidelines in Percie du Sert et al [2020])

This study compared *Npc1*$^{m1N/m1N}$ mice administered either vehicle (saline) or gene therapy vector (AAV9 or AAV-PHP.B) with reporter construct (GFP) or *NPC1*. *Npc1*$^{+/+}$ mice included as a control group for behavioral, survival, and pathology analyses did not receive saline, whereas a subset of *Npc1*$^{+/+}$ mice received either the AAV9-GFP or AAV9-*NPC1* for biodistrubution and gene copy number analyses. The *Npc1*$^{m1N/m1N}$ colony has been maintained in-house for 5+ yr but originated from the Jackson Laboratory (Stock #003092, BALB/cNctr-Npc1$^{m1N}$/J). Each mouse was a single experimental unit and the sample size was based on previously published work from our group. A total of 82 mice were used in these studies (Table S2). No explicit criteria were set a priori for inclusion/exclusion criteria and no mice or data points were excluded from study or analyses. Group sample size for each analysis is stated in text, figure legend, or figure. Randomization was employed by using multiple cohorts and mice within each treatment group were included in every cohort to minimize confounders. In addition, order of mice in behavioral tests varied with each testing date and all animals were group housed. Researchers were not blinded to treatment at time

of injections. However, behavioral analyses were carried out in a blinded fashion such that colored tails were used to denote individual mice within cages and only the cage number and tail color were available to the evaluator during testing. Main outcome measures, provided throughout text, included survival, behavioral phenotype, pathology, and gene copy number. Statistical test selection was based on particular data set and accounted for normalized (or not) data.

## Behavior testing

Two behavioral assays were used to determine the effect of gene therapy on motor performance: phenotype score and balance beam. Mice were tested beginning at 6 wk of age and then every 3 wk thereafter until euthanasia or unable to complete the task. The phenotype score evaluates six individual domains associated with the disease phenotype seen in $Npc1^{m1N/m1N}$ mice: gait, kyphosis, ledge test, and hind limb clasps from a cerebellar ataxia score (Guyenet et al, 2010) plus grooming and tremor (Alam et al, 2016). Each domain is given a score of 0–3 with a higher score indicating greater disease progression and the composite score of all domains is presented in results. The balance beam assay is a quantitative approach for assessing loss of motor function (Gulinello et al, 2010; Arteaga-Bracho et al, 2016). The number of hind limb foot slips is counted as mice traverse a four-foot long, round wooden beam (diameter = 18 or 24 mm). A more progressed disease state correlates to a higher number of slips.

## Administration of vector

$Npc1^{m1N/m1N}$ mice received a 50 $\mu$l retro-orbital injection at weaning (24–27 d) of one of the following: $1.43 \times 10^{12}$ GC of AAV-PHP.B-EF1a(s)-$NPC1$ (n = 13), $1.84 \times 10^{12}$ GC of AAV9-EF1a(s)-$NPC1$ (n = 12), $1.21 \times 10^{12}$ GC of AAV-PHP.B-EF1a(s)-GFP (n = 2), or $1.21 \times 10^{12}$ GC of AAV9-EF1a(s)-GFP (n = 2). Control $Npc1^{m1N/m1N}$ mice received a 50-$\mu$l retro-orbital injection of 0.9% saline at weaning (24–27 d).

## Tissue collection and homogenization

Euthanasia for tissue collection was initiated with an intraperitoneal injection of Avertin (lethal dose of 0.04 ml/gm). When mice were insensate, the chest cavity was opened to allow a transcardiac perfusion of 0.9% saline. One half of the brain and a piece of liver were then collected and flash frozen in liquid nitrogen, with long-term storage at –80°C. Subsequently, mice were re-perfused with 4% paraformaldehyde before collecting remaining organs for post-fixation overnight in 4% paraformaldehyde. Fixed tissues were rinsed and stored in PBS at 4°C.

Tissue homogenization was achieved using a Benchmark Scientific BeadBug homogenizer and UltraPure $H_2O$ (11005-060; IPM Scientific). Frozen tissues were placed in tubes prefilled with 3 mm glass beads and homogenized 3 × 30 s at a speed of 400. Homogenate was immediately aliquoted into three separate tubes for DNA (ddPCR), metabolomics, and protein (Western blot; WB). RIPA buffer with proteinase inhibitor cocktail (20-201; Millipore) was added to WB samples and then homogenates were spun down at

14,534$g$ for 20 min at 4°C. Supernatant was collected and stored at –80°C.

## Western blotting

Protein levels from cerebrum and liver WB homogenates were quantified using the BCA Protein Assay kit from Pierce (23227). Equal amounts of protein (80 $\mu$g for liver and 120 $\mu$g for cerebrum) were run on 4–12% Bis-Tris SDS-polyacrylamide gels (NW04120BOX; Thermo Fisher Scientific) to achieve separation of protein bands. After transferring to a nitrocellulose membrane (IB301002; Life Technologies) and blocking for 1 h in TBS-Tween + LI-COR Odyssey Blocking Buffer (927-40000), samples were incubated overnight with two antibodies: rabbit anti-NPC1 (ab 134113; 1:1,000; Abcam) and the loading control mouse anti-$\alpha$-tubulin (T9026; 1:1,000; Millipore). The Odyssey donkey anti-rabbit 680 (926-68073; 1:5,000; LI-COR Biosciences) and the Odyssey donkey anti-mouse 800 (926-32212; 1:5,000; LI-COR Biosciences) were used as secondary antibodies. The LI-COR Odyssey Imaging System was used to capture results.

## Immunohistochemistry

Brain and liver tissue from each treatment group at 9-wk-old and end stage were used for immunohistochemical staining. 24–48 h before sectioning of GFP biodistribution samples, tissues were transferred to a 30% sucrose solution where they remained until sinking. Using a cryostat, brain and liver were sectioned para-sagittally at a 30 $\mu$m thickness and free-floating sections were collected and washed in PBS containing 0.25% Triton X-100 (PBSt). After a 1-h block at room temperature in PBSt/normal goat serum (NGS, S26-M; Sigma-Aldrich), primary antibodies were diluted in PBSt/NGS and sections incubated overnight. Subsequent to washing in PBSt, appropriate secondary antibodies were diluted in PBSt/NGS and incubated with sections for 30 min at room temperature. Refer to Table S2 for antibodies and dilutions. Filipin complex from *Streptomyces filipinensis* (F9765; 0.025 g/ml; Sigma-Aldrich) was diluted in PBSt to stain tissue for cholesterol accumulation. ProLong Gold mounting medium with or without DAPI (P36930 or P36935; Life Technologies) was used to coverslip slides after sections were mounted. H&E staining was performed by Histoserv, Inc..

## Quantification of CD68⁺ and IBA1⁺ area

Percent CD68⁺ area relative to total area in liver tissue sections was performed according the methods previously described (Rodriguez-Gil et al, 2020). Percent IBA1+ area relative to total area in cerebellar tissue sections was performed in a similar manner with the use of a $\beta$ version of Image-Pro v 10.5 software (Media Cybernetics, Inc.).

## Copy number analysis by ddPCR

ddPCR was performed as previously described (Lissa et al, 2018) with the following modifications: NPC1 and GAPDH primers were obtained from Bio-Rad (unique Assay IDs dCNS361140976 with 6-FAM and dMmuCNS133125454 with HEX, respectively). From sample

homogenates, 50 ng of DNA from cerebrum or cerebellum or 10 ng of DNA from liver was used to quantify gene copy number. PCR cycling conditions were identical with the exception of 60°C for annealing and extension. Droplet signal was read as being either positive or negative for NPC1 and/or GAPDH. Any samples with fewer than 10,000 positive droplets were excluded and the sample(s) re-run to obtain an accurate read.

### Ly6a genotype analysis

Genomic DNA was amplified with primers and probes using a real-time SNP genotyping assay (Custom Taqman Assay Design Tool; Thermo Fisher Scientific) for SNPs rs32279213 and rs213983347, as follows: rs32279213 F: GCAGATGGGTAAGCAAAGATTGTTC, R: GTCCCTG CATAAGAAGTGAGTCA, FAM:TTCTTGCAGGTTCTCA,VIC: TTTCTTGCAGATT CTCA, rs213983347: F: AGGTGCTGCCTCCATTGG, R: CTAAGGTCAACGTGA AGACTTCCT, FAM: TCTGCAATGCAGCAGT, VIC: CCTCTGCAATGTAGCAGT, a universal 2x Taqman Master Mix (Thermo Fisher Scientific) and an ABI 7500 instrument for thermocycling and detection. For the genome-wide scan of two $Npc1^{m1N/m1N}$ mice, data were assessed at the DartMouseTM Speed Congenic Core Facility at the Geisel School of Medicine at Dartmouth, using a custom panel of 5,307 SNPs distributed throughout the mouse genome. Raw SNP data were analyzed using DartMouse's SNaP-MapTM and Map-SynthTM software.

### Lipidomics

Lipidomic analyses on cerebral, cerebellar, and liver homogenates were carried out as previously described (Praggastis et al, 2015; Davidson et al, 2019).

### Image capture and analysis

Fluorescent images were captured on an inverted Zeiss Axio Observer.Z1 using an AxioCam MRm and ZEN 2.5 software. Bright-field images were captured on an inverted Zeiss Axio Observer.D1 using an AxioCamHRc and ZEN 2011 software. Adobe Photoshop 2020 version 21.1.2 was used to resize and adjust brightness and contrast, such that all images within a staining run and/or figure were modified in an identical manner.

### Statistical analysis

Statistical analyses were done using GraphPad Prism version 8.0.0 for Windows or Mac (GraphPad Software, San Diego, California, USA, www.graphpad.com). The following statistical tests were used: Mantel–Cox log rank test (Fig 2A); Kruskal–Wallis test with Dunn's multiple comparisons test (Figs 2C and S7); Welch's ANOVA test with Dunnett's multiple comparisons test (Fig 2D); two-way ANOVA with Tukey's multiple comparisons test (Figs 2E and S3); unpaired $t$ test (Figs 3A–C and S4); and Spearman's correlation coefficient test (Figs 3D–F and S2). For data sets, normality was evaluated if possible, and if not, a non-parametric test was used. Otherwise, the appropriate statistical test was then selected for further analysis. All data is presented as mean ± SD unless otherwise indicated.

### Study approval

All animal works were performed according to National Institutes of Health–approved animal care and use protocols.

## Supplementary Information

## Acknowledgements

We are grateful for the wealth of expertise and technical assistance regarding droplet digital PCR provided by Valery Bliskovsky, Steven Shema, and Liz Conner in the National Cancer Institute's Genomics Core. The dedicated National Institutes of Health (NIH) animal care and veterinary staff is gratefully acknowledged for their commitment to the care and well-being of all animals in our facilities, especially the mice included in this study. Finally, and perhaps most importantly, the unwavering support and trust in research by NPC-affected patients and families is a powerful reminder of why our collective efforts as a research community must prevail. This research was funded by the Intramural Research Program of the National Human Genome Research Institute (NHGRI) at the NIH (1ZIAHG000068-15). Additional support was generously provided by Liferay, Inc. (Postbaccalaureate Intramural Research Training Award for AL Gibson) and Niemann-Pick Canada and Ara Parseghian Medical Research Fund at the University of Notre Dame. Support for CD Davidson was kindly provided by the Hide & Seek Foundation and Dana's Angels Research Trust (both part of Support Of Accelerated Research for Niemann-Pick C). Work performed at the California Institute of Technology was supported by NIH Pioneer grant (DP1OD025535; V Gradinaru) and the Beckman Institute for CLARITY, Optogenetics and Vector Engineering Research (CLOVER) for technology development and dissemination (V Gradinaru). JL Rodriguez-Gil was supported by an NHGRI Intramural Research Training Award, the NIH Oxford-Cambridge Scholars Program, and the Medical Scientist Training Program from the School of Medicine and Public Health, University of Wisconsin-Madison (3T32GM008692).

### Author Contributions

CD Davidson: conceptualization, formal analysis, investigation, visualization, project administration, and writing—original draft, review, and editing.
AL Gibson: investigation and writing—original draft, review, and editing.
T Gu: investigation and writing—original draft, review, and editing.
LL Baxter: formal analysis, visualization, and writing—original draft, review, and editing.
BE Deverman: conceptualization, resources, investigation, and writing—review and editing.
K Beadle: investigation and writing—review and editing.
AA Incao: investigation and writing—review and editing.
JL Rodriguez-Gil: formal analysis, investigation, and writing—review and editing.
H Fujiwara: formal analysis and writing—review and editing.
X Jiang: formal analysis, investigation, and writing—review and editing.
RJ Chandler: conceptualization and writing—review and editing.
DS Ory: resources and writing—review and editing.

V Gradinaru: resources, funding acquisition, and writing—review and editing.

CP Venditti: conceptualization, resources, supervision, funding acquisition, and writing—review and editing.

WJ Pavan: conceptualization, resources, supervision, funding acquisition, project administration, and writing—review and editing.

## Conflict of Interest Statement

The NIH has filed patents on behalf of RJ Chandler, CP Venditti, and WJ Pavan related to NPC1 gene therapy and biomarkers. The California Institute of Technology has filed patents related to AAV-PHP.B with BE Deverman and V Gradinaru as inventors. BE Deverman also has a patent application filed by the Broad Institute and is a consultant for Voyager Therapeutics.

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
