## [Reviewer comments · Life Science Alliance]

Life Science Alliance

Improved systemic AAV gene therapy with a neurotrophic capsid in Niemann-Pick disease type C1 mice

Cristin Davidson, Alana Gibson, Tansy Gu, Laura Baxter, Benjamin Deverman, Keith Beadle, Arturo Incao, Jorge Rodriguez-Gil, Hideji Fujiwara, Xuntian Jiang, Randy Chandler, Daniel Ory, Viviana Gradinaru, Charles Venditti, and William Pavan

DOI: <https://doi.org/10.26508/lsa.202101040>

Corresponding author(s): William Pavan, National Human Genome Research Institute and Charles Venditti, NIH

Review Timeline:

Submission Date:	2021-01-28
Editorial Decision:	2021-03-25
Revision Received:	2021-07-27
Editorial Decision:	2021-07-29
Revision Received:	2021-08-04
Accepted:	2021-08-04

Transaction Report:

March 25, 2021

Re: Life Science Alliance manuscript #LSA-2021-01040

Dr. Bill Pavan
National Human Genome Research Institute
Genetic Disease Research Branch Building 49, Room 4A82 49 Convent Drive, MSC 4472
Bethesda, MD 20892-4472

Dear Dr. Pavan,

Thank you for submitting your manuscript entitled "A neurotrophic capsid improves the efficacy of systemic AAV gene therapy in Niemann-Pick C1 mice" to Life Science Alliance. The manuscript was assessed by expert reviewers, whose comments are appended to this letter.

We apologize for this unusual and extended delay in getting back to you. As you can see from the reviewers' comments below, both reviewers are interested in these findings. Reviewer 1 has only a few points, while Reviewer 2 has raised a number of questions, most of which, in our opinion, could be addressed with text revisions and reasonable experimentation. We would, thus, encourage you to submit a revised version of the manuscript that addresses all of the reviewers' points.

Thank you for this interesting contribution to Life Science Alliance. We are looking forward to receiving your revised manuscript.

Sincerely,

Shachi Bhatt, Ph.D.

Executive Editor

Life Science Alliance

<https://www.lsjournal.org/>

Interested in an editorial career? EMBO Solutions is hiring a Scientific Editor to join the international Life Science Alliance team. Find out more here -

https://www.embo.org/documents/jobs/Vacancy_Notice_Scientific_editor_LSA.pdf

B. MANUSCRIPT ORGANIZATION AND FORMATTING:

Reviewer #1 (Comments to the Authors (Required)):

This manuscript reports on the consequences of gene therapy in a mouse model of NPC disease, comparing two viral vectors that show preferential transduction across the BBB versus to the liver. The manuscript is well written and the findings are of broad importance; the authors show that even modest gene rescue has significant benefit for the mice. The story should be published without delay and the following comments are offered to the authors to enhance presentation and

clarity for the reader.

1. Supplemental Figure 5 please state more clearly in the text how to interpret the color pattern to facilitate interpretation for the reader. Please comment specifically on changes in cholesterol in the 3 tissues.

2. Figure 1 and 5. Is there any way to quantify image intensity over particular areas to permit quantitative comparison (using Image J for example?) This should be easy and would be of value to capture the impact of the images shown.

Overall, well done! -Suzanne Pfeffer (signed review)

Reviewer #2 (Comments to the Authors (Required)):

In this manuscript, Davidson et al report a gene therapy approach for NPC1 disease which is a fatal neurodegenerative disease. In the study, they compare both AAV9 and AAVPHP-B-NPC1 for their therapeutic efficiency after intravenous delivery in the mouse model of the disease. They demonstrate a quite good, survival and weight improvement in AAVPHPB NPC1 treated mice compare to the AAV9 treated one.

The manuscript is overall well written and conclusion in accordance to the described data.

However, I have this major points to highlight:

- In the first study they are comparing both vector using a GP reporter gene with a dose of 1.21×10^{12} vg. Could you please comment on how this dose was decided?
- Then in the treatment study they are using other doses without any special comment and more strikingly, AAV9 and AAVPHPB groups did not receive the same amount of vg. Could authors comment why they use other dose and especially why the 2 groups were not injected with the same vg to be able to really compare treatment efficacy.
- To assess effect of treatment on behavior of NPC1 mice, gait analysis or maybe clasping test would have been appreciated. Could authors comment on why they decide to use their phenotype score and beam walk test, which seems not the best appropriated to the model.
- Regarding VGC determination; compare to the image shown in Figure 1, VGC for AAVPHPB is not so high in cerebrum, could authors detail the VGC in several brain area (hippocampus, cortex....). And what about other peripheral organs, did VGC was assessed and why showing only liver?
- In AAV9 treated mice, authors showed a number of 80VGC, no comments are done on a potential toxicity and it is difficult to really compare with AAVPHPB as the total injected dose was not the same.
- In figure 4, did expression of NPC1 in liver and cerebrum was evaluated in parallel as AAV9 liver display 80VGC and AAVPHPB cerebrum around 2-3 VGC however, the levels of expression seems similar, which is quite puzzling, Could the authors comment on that ?
- All the part regarding Ly6 receptor is quite disturbing for me, indeed it is known and published that at least in mice AAVPHPB need the Ly6 receptor to transduce the CNS, so why some mice in which My6 was not present were used for the study, a first screening of their mice expressing the Ly6 receptor should have been done to only use them for the study, could the authors comment on that ? this for me is more supplementary data
- Finally, Iba1 staining is not convincing for me even in NPC1 saline mice, could authors presents a better staining for that
- For liver analysis, no comments on AAV9 toxicity related, and is not clear for me how a 80VGC transduction would only need to a modest improvement. Could the authors comment on that

Minor points:

- For figure 2C and D, the normal growth curve which is present in supplementary seems more clear than the 2 extrapolate graphs

Response to Reviewers

We appreciate the thoughtful comments from reviewers and welcome the opportunity to further clarify and enhance our manuscript based on their suggestions. Please find our point by point responses below.

From Reviewer #1:

1. Supplemental Figure 5. Please state more clearly in the text how to interpret the color pattern to facilitate interpretation for the reader. Please comment specifically on changes in cholesterol in the 3 tissues.
 - Thank you for pointing out that this figure needed clarification. We have included the following statement in the legend for Supplemental Figure 5 (p. 50, lines 965-967): “The colorimetric scale (lower right) reflects the minimum and maximum levels of individual lipids in each row, with green indicating lowest levels, red indicating highest levels, and black indicating intermediate levels.” We have also added additional phrases mentioning the green and red colors and noting these mean lower and higher expression levels, respectively (p. 50).
 - Text has been added to p. 14, lines 257-259, to describe the absence of consistent cholesterol changes: “No consistent changes in cholesterol levels were apparent in brain or liver tissues of AAV9-*NPC1*- and AAV-PHP.B-*NPC1*-treated *Npc1*^{m1N/m1N} mice.”
2. Figure 1 and 5. Is there any way to quantify image intensity over particular areas to permit quantitative comparison (using Image J for example?). This should be easy and would be of value to capture the impact of the images shown.
 - We acknowledge the reviewer’s desire to have quantified biodistribution data, but we believe this is not critical to the conclusions drawn from our work. Figure 1 confirms previous studies by Deverman et al. (Nature Biotechnology, 2016), which showed that the AAV-PHP.B vector has greater CNS transduction than AAV9. Deverman’s work quantified transduction differences between the vectors in brain and several other organs. Our qualitative results support this conclusion by clearly demonstrating greater AAV-PHP.B-GFP expression compared to AAV9-GFP in the *Npc1* mouse model.
 - With respect to the filipin labeling of unesterified cholesterol in Figure 5: Regretfully, we are unable to stain additional sections of brain necessary for filipin quantification since all tissues have been embedded in paraffin for subsequent processing. The embedding process removes lipids rendering filipin and other lipid labels useless. To clarify that we do not conclude one vector has superior amelioration of brain pathology, we highlight the heterogeneity seen with both vectors in terms of effect on unesterified cholesterol in the CNS in the legend of Figure 5 (p. 43, line 911): “Note heterogeneity of modest correction observed in both AAV9-*NPC1* and AAV-PHP.B-*NPC1* transduced mice (C-F).”
 - We were, however, able to stain additional brain sections and quantify the percent area of anti-IBA1 labeling relative to total area in specific cerebellar lobules. This represents a direct quantification of microglia as requested by Reviewer 1. The new IBA1 quantification is included in Supplemental Figure 7 as well as in the text, p. 16, lines 291-304: “Quantification of microgliosis, or the percentage of IBA1⁺ area, was determined in lobules III, VI/VII, and IX of cerebellar tissue sections. Saline-injected *Npc1*^{m1N/m1N} mice displayed a significantly higher percentage IBA1⁺ area as compared to *Npc1*^{+/+} mice (Supplemental Table 2; Kruskal-Wallis test with Dunn’s multiple comparison post-test, $P=.0029$ for lobule III, $P=.0034$ for lobules VI/VII, $P=.0264$ for lobule IX). IBA1⁺ area in specified lobules of cerebella from *Npc1*^{m1N/m1N} mice treated with either gene therapy vector were not significantly different from saline-injected *Npc1*^{m1N/m1N} or *Npc1*^{+/+} mice, though AAV9-*PHP.B-NPC1*-treated *Npc1*^{m1N/m1N} mice did trend toward greater reduction in pathology. This finding correlates with the higher AAV9-*PHP.B-NPC1* vector copy number and *NPC1* protein levels in brain.”

From Reviewer #2:

1. In the first study they are comparing both vector using a GP reporter gene with a dose of 1.21E12vg. Could you please comment on how this dose was decide?
 - A dose of 1.21E12 vg/mouse was selected based on our previous publication showing efficacy of AAV9 (Chandler et al, 2016).
2. Then in the treatment study they are using other doses without any special comment and more strikingly, AAV9 and AAVPHPB groups did not receive the same amount of vg. Could authors comments why they use other dose and especially why the 2 groups were not injected with the same vg to be able to really compare treatment efficacy.
 - We appreciate the reviewer noting these differences in vector dosages and provide the following explanation. Very shortly after the publication highlighting AAV-PHP.B's superior CNS transduction, we initiated a collaboration between the National Human Genome Research Institute and the California Institute of Technology. This was very early in their production process and due to technical issues with the titer assay, the final titers were slightly different than those used for calculations during dilution for retro-orbital administration. The technical difficulty with titer resulted in delivery of a ~22% higher dose of the AAV9-*NPCI* compared to AAV-PHP.B-*NPCI*. While this difference was not intentional, it does err in favor of AAV9 and, in essence, further validates the finding that AAV-PHP.B has greater CNS transduction than AAV9.
 - Text has been added on p. 9, lines 138-139 to clarify this issue: "Technical variability with vector titer assays led to these moderately different doses for AAV-PHP.B-*NPCI* vs AAV9-*NPCI*."
3. To assess effect of treatment on behavior of NPC1 mice, gait analysis or maybe clasping test would have been appreciated. Could authors comment on why they decide to use their phenotype score and beam walk test, which seems not the best appropriated to the model.
 - We agree with the reviewer that both gait and hindlimb clasp can be very useful measures for NPC mouse models. In fact, both parameters are included in the composite phenotype score. For clarity, we have updated the results text to include the 6 measures of the composite score (p. 10, lines 164-165), along with descriptions in the Materials and Methods (p. 24, line 469) and legend for Figure 2 (p. 40, lines 869-870). We feel that the composite score – which additionally includes grooming, kyphosis, tremor, and ledge test – is a good readout measure that encompasses more of the overall phenotype. While acknowledging that objective quantitative data is preferred, we do not have access to specialized systems such as the Noldus CatWalk XT and hence the balance beam is a rapid, low cost alternative.
4. Regarding VGC determination; compare to the image shown in Figure 1, VGC for AAVPHPB is not so high in cerebrum, could authors detail the VGC in several brain area (hippocampus, cortex,...). And what about other peripheral organs, did VGC was assessed and why showing only liver?
 - All frozen samples collected from the study mice and subsequently used for gene copy number analysis have been homogenized. The brain was collected and homogenized in two parts: cerebellum and cerebrum (which includes areas such as hippocampus, cortex, and thalamus), thus brain sub-regions from these mice are not available.
 - Our experimental design involved collection of only brain and liver for gene copy number analysis, due to the large number of mice in the study. We selected these two organs because they both have a well-documented role in NPC disease, opting to further isolate the cerebellum due to the significant impact of Purkinje neuron loss on phenotypic manifestations.

5. In AAV9 treated mice, authors showed a number of 80VGC, no comments are done on a potential toxicity and it is difficult to really compare with AAVPHPB s the total injected dose was not the same.
 - We acknowledge that the doses for AAV9-*NPC1* and AAV-PHP.B-*NPC1* differ due to technical issues described in point 2, with AAV9 being administered at a higher dose. As mentioned above in point 2, this difference has been noted in the manuscript (p. 9, lines 138-139). While toxicity is possible with AAV9, we found no histological indicators consistent with hepatic injury or genotoxicity. Furthermore, the mice had no adverse clinical symptoms as have been described in NHPs or piglets treated with high doses of AAV9 early in life (Hinderer et al., 2018). Upon examination of H&E staining for liver in *Npc1*^{m1N/m1N} mice treated with AAV9-*NPC1*, the cytoarchitecture actually looked very similar to *Npc1*^{+/+} liver and showed no obvious signs of toxicity (Fig. 6 and Supplemental Fig. 8). Text describing these observations is now included in the results (p. 17-18, lines 331-338).
 - We have performed an additional analysis on the liver, quantification of the percent area positive for macrophages (Kuppfer cells) as assessed by CD68 labeling. No statistically significant differences were found between the AAV9-*NPC1* and AAV-PHP.B-*NPC1* treated mouse livers. In fact, the only significant difference was observed between *Npc1*^{m1N/m1N} mice administered saline and the *Npc1*^{+/+} mice, an expected finding.
 - Text has been added to p.16, lines 323-330: “Finally, the percentage of CD68⁺ area in sectioned liver tissue was compared between the different treatment groups (Supplemental Fig.7B). Livers from saline-injected *Npc1*^{m1N/m1N} mice displayed significantly larger CD68⁺ areas than did livers from *Npc1*^{+/+} mice (Kruskal-Wallis test with Dunn’s multiple comparison post-test; $P=.0109$). Although neither AAV9-*NPC1*- nor AAV-PHP.B-*NPC1*-treated *Npc1*^{m1N/m1N} mice showed significant differences from values for *Npc1*^{m1N/m1N} mice administered saline, there was a trend toward reduced CD68⁺ area in the AAV9-*NPC1*-treated mice which correlates with the assessments of vector copy number and NPC1 protein levels in liver.” Additionally, Supplemental Fig. 7B has been added with mean, standard deviation, and standard error of mean for the assessment. The reference Rodriguez-Gil et al, 2020 has been cited for method of quantification.
6. In figure 4, did expression of NPC1 in liver and cerebrum was evaluated in parallel as AAV9 liver display 80VGC and AAVPHPB cerebrum around 2-3 VgC however, the levels of expression seem similar, which is quite puzzling, Could the authors comment on that?
 - A couple of factors explain why the Western blot bands for NPC1 in liver and cerebrum appear similar. First, in order to detect NPC1 expression in the cerebrum, it was necessary to load 120 µg of protein vs. only 80 µg of protein for liver. Second, the acquisition settings for liver and cerebrum blots were configured independently to allow detection of bands within homogenate of either organ. A longer exposure time and higher gain were required to detect very faint bands in cerebrum (as evidenced by the over-exposed ladder in cerebrum blot). For liver, NPC1 expression was much stronger and required lower gain and exposure to visualize expression (again, lighter ladder indicates lower settings).
7. All the part regarding Ly6a receptor is quite disturbing for me, indeed it is known and published that at least in mice AAVPHPB need the Ly6a receptor to transduce the CNS, so why some mice in which My6 was not present where used for the study, a first screening of their mice expressing the Ly6 receptor should have been done to only use them for the study, could the authors comments on that? This for me is more supplementary data
 - Thank you for pointing out this important concern. Our study evaluating AAV-PHP.B in NPC disease commenced in 2016, but the first work documenting the Ly6a dependence of AAV-PHP.B in mice was not published until 2019 (Hordeaux et al, 2019), three years following the initiation of our study. Therefore, we were unaware of this confounding issue until all of our mice had already been injected and sacrificed. We have edited text on p. 12,

lines 215-217 to clarify this accordingly: “Following the completion of the AAV-PHP.B-*NPC1* experiments, other groups published work identifying strain-specific effects on the CNS transduction efficiency of AAV-PHP.B that are associated with different haplotypes at *Ly6a*...”

8. Finally, Iba1 staining is not convincing for me even in NPC1 saline mice, could authors presents a better staining for that
 - We appreciate the constructive feedback related to immunofluorescence labeling of IBA1. We have performed additional staining and quantification of the percent IBA1⁺ area relative to total region of interest. This data is included in Supplemental Fig. 7 and in the text on p. 16, lines 291-304: “Only modest improvement of gliosis in gene therapy treated mice compared to the saline-injected *Npc1*^{m1N/m1N} cohort was observed, as evidenced by microglial and astrocytic staining (anti-IBA1 and anti-GFAP, respectively). Quantification of microgliosis, or the percentage of IBA1⁺ area, was determined in lobules III, VI/VII, and IX of cerebellar tissue sections. Saline-injected *Npc1*^{m1N/m1N} mice displayed a significantly higher percentage IBA1⁺ area as compared to *Npc1*^{+/+} mice (Supplemental Fig. 7A; Kruskal-Wallis test with Dunn’s multiple comparison post-test, *P*=.0029 for lobule III, *P*=.0034 for lobules VI/VII, *P*=.0264 for lobule IX). IBA1⁺ area in specified lobules of cerebella from *Npc1*^{m1N/m1N} mice treated with either gene therapy vector were not significantly different from saline-injected *Npc1*^{m1N/m1N} or *Npc1*^{+/+} mice, though AAV9-PHP.B-*NPC1*-treated *Npc1*^{m1N/m1N} mice did trend toward greater reduction in pathology. This finding correlates with the higher AAV9-PHP.B-*NPC1* vector copy number and NPC1 protein levels in brain.”
9. For liver analysis, no comments on AAV9 toxicity related, and is not clear for me how a 80 VCG transduction would only need to a modest improvement. Could the authors comment on that
 - The revised manuscript now indicates that there were no signs of AAV9 toxicity, as described in point 5 above.
 - We agree that an average of 80 vector copies per cell seen in livers of AAV9-*NPC1* treated *Npc1*^{m1N/m1N} mice could suggest that liver pathology should be greatly reduced. As with any assay, there are limitations to the droplet digital PCR assay. In particular, two scenarios may account for higher copy numbers: a greater number of cells may have a lower copy number per cell *or* a smaller number of cells may have a higher copy number per cell. We cannot conclude which scenario best represents of our data.
10. For figure 2C and D, the normal growth curve which is present in supplementary seems more clear than the 2 extrapolate graphs
 - We agree that the weight curves are quite easy to visualize, however, they may be misinterpreted since mice continually drop out of the curve upon humane sacrifice. Therefore, weekly weights are not available for every mouse because of the different ages of sacrifice in the survival cohort. In contrast, the graph in 2C highlights when each mouse reached its peak weight. This represents a single time point per mouse and is available for every mouse in the study.
 - Graph 2D provides a single measurement of weight gain, loss, or maintenance for every mouse from 6-9 weeks of age, including *Npc1*^{+/+} mice. It is well documented that untreated *Npc1*^{m1N/m1N} mice experience a precipitous decline in weight during this time period. Successful therapeutic interventions often delay this phenotypic manifestation. Therefore, this measurement provides a quantifiable and easily visualized way to identify a positive impact.
 - For the reasons outlined above, we believe the weight curves should remain as supplemental figures.

July 29, 2021

RE: Life Science Alliance Manuscript #LSA-2021-01040R

Dr. William J. Pavan
National Human Genome Research Institute
Genetic Disease Research Branch Building 49, Room 4A82 49 Convent Drive, MSC 4472
Bethesda, MD 20892-4472

Dear Dr. Pavan,

Thank you for submitting your revised manuscript entitled "Improved systemic AAV gene therapy with a neurotrophic capsid in Niemann-Pick disease type C1 mice". We would be happy to publish your paper in Life Science Alliance pending final revisions necessary to meet our formatting guidelines.

- please add ORCID ID for the corresponding (and secondary corresponding) author-both of you should have received instructions on how to do so
- please add the Twitter handle of your host institute/organization as well as your own or one of the first author in our system
- please check whether all Authors are inserted in the Author Contributions section of your main manuscript text
- please add callouts for Figures 5B and S6A-H to your main manuscript text

FIGURE CHECKS:

- there are panels A-X in Figure S8, but they are not mentioned in the figure legend nor are there callouts for them in the manuscript text
- same issue as above for Figure S1, A, and B panels

LSA now encourages authors to provide a 30-60 second video where the study is briefly explained. We will use these videos on social media to promote the published paper and the presenting author. Corresponding or first-authors are welcome to submit the video. Please submit only one video per manuscript. The video can be emailed to contact@life-science-alliance.org

A. FINAL FILES:

B. MANUSCRIPT ORGANIZATION AND FORMATTING:

Sincerely,

Reviewer #1 (Comments to the Authors (Required)):

This authors have provided thoughtful and suitable responses to the reviewer comments and the paper should be published without delay. They should note that image quantitation would not require new section labeling and would rely on the images already obtained.

Reviewer #2 (Comments to the Authors (Required)):

I thanks the reviewer for their reply and answering all the pointed out item and the paper is now fine for publication

Davidson et al., AAV9 vs AAV-PHP.B in NPC disease
LSA-2021-01040R
7.26.2021

Response to Reviewers

We appreciate the thoughtful comments from reviewers and welcome the opportunity to further clarify and enhance our manuscript based on their suggestions. Please find our point by point responses below.

From Reviewer #1:

1. Supplemental Figure 5. Please state more clearly in the text how to interpret the color pattern to facilitate interpretation for the reader. Please comment specifically on changes in cholesterol in the 3 tissues.
 - Thank you for pointing out that this figure needed clarification. We have included the following statement in the legend for Supplemental Figure 5 (p. 50, lines 965-967): “The colorimetric scale (lower right) reflects the minimum and maximum levels of individual lipids in each row, with green indicating lowest levels, red indicating highest levels, and black indicating intermediate levels.” We have also added additional phrases mentioning the green and red colors and noting these mean lower and higher expression levels, respectively (p. 50).
 - Text has been added to p. 14, lines 257-259, to describe the absence of consistent cholesterol changes: “No consistent changes in cholesterol levels were apparent in brain or liver tissues of AAV9-*NPC1*- and AAV-PHP.B-*NPC1*-treated *Npc1*^{m1N/m1N} mice.”
2. Figure 1 and 5. Is there any way to quantify image intensity over particular areas to permit quantitative comparison (using Image J for example?). This should be easy and would be of value to capture the impact of the images shown.
 - We acknowledge the reviewer’s desire to have quantified biodistribution data, but we believe this is not critical to the conclusions drawn from our work. Figure 1 confirms previous studies by Deverman et al. (Nature Biotechnology, 2016), which showed that the AAV-PHP.B vector has greater CNS transduction than AAV9. Deverman’s work quantified transduction differences between the vectors in brain and several other organs. Our qualitative results support this conclusion by clearly demonstrating greater AAV-PHP.B-GFP expression compared to AAV9-GFP in the *Npc1* mouse model.
 - With respect to the filipin labeling of unesterified cholesterol in Figure 5: Regretfully, we are unable to stain additional sections of brain necessary for filipin quantification since all tissues have been embedded in paraffin for subsequent processing. The embedding process removes lipids rendering filipin and other lipid labels useless. To clarify that we do not conclude one vector has superior amelioration of brain pathology, we highlight the heterogeneity seen with both vectors in terms of effect on unesterified cholesterol in the CNS in the legend of Figure 5 (p. 43, line 911): “Note heterogeneity of modest correction observed in both AAV9-*NPC1* and AAV-PHP.B-*NPC1* transduced mice (C-F).”
 - We were, however, able to stain additional brain sections and quantify the percent area of anti-IBA1 labeling relative to total area in specific cerebellar lobules. This represents a direct quantification of microglia as requested by Reviewer 1. The new IBA1 quantification is included in Supplemental Figure 7 as well as in the text, p. 16, lines 291-304: “Quantification of microgliosis, or the percentage of IBA1⁺ area, was determined in lobules III, VI/VII, and IX of cerebellar tissue sections. Saline-injected *Npc1*^{m1N/m1N} mice displayed a significantly higher percentage IBA1⁺ area as compared to *Npc1*^{+/+} mice (Supplemental Table 2; Kruskal-Wallis test with Dunn’s multiple comparison post-test, *P*=.0029 for lobule III, *P*=.0034 for lobules VI/VII, *P*=.0264 for lobule IX). IBA1⁺ area in specified lobules of cerebella from *Npc1*^{m1N/m1N} mice treated with either gene therapy vector were not significantly different from saline-injected *Npc1*^{m1N/m1N} or *Npc1*^{+/+} mice, though AAV9-*PHP.B-NPC1*-treated *Npc1*^{m1N/m1N} mice did trend toward greater reduction in pathology. This finding correlates with the higher AAV9-*PHP.B-NPC1* vector copy number and *NPC1* protein levels in brain.”

From Reviewer #2:

1. In the first study they are comparing both vector using a GP reporter gene with a dose of 1.21E12vg. Could you please comment on how this dose was decide?
 - A dose of 1.21E12 vg/mouse was selected based on our previous publication showing efficacy of AAV9 (Chandler et al, 2016).
2. Then in the treatment study they are using other doses without any special comment and more strikingly, AAV9 and AAVPHPB groups did not receive the same amount of vg. Could authors comments why they use other dose and especially why the 2 groups were not injected with the same vg to be able to really compare treatment efficacy.
 - We appreciate the reviewer noting these differences in vector dosages and provide the following explanation. Very shortly after the publication highlighting AAV-PHP.B's superior CNS transduction, we initiated a collaboration between the National Human Genome Research Institute and the California Institute of Technology. This was very early in their production process and due to technical issues with the titer assay, the final titers were slightly different than those used for calculations during dilution for retro-orbital administration. The technical difficulty with titer resulted in delivery of a ~22% higher dose of the AAV9-*NPCI* compared to AAV-PHP.B-*NPCI*. While this difference was not intentional, it does err in favor of AAV9 and, in essence, further validates the finding that AAV-PHP.B has greater CNS transduction than AAV9.
 - Text has been added on p. 9, lines 138-139 to clarify this issue: "Technical variability with vector titer assays led to these moderately different doses for AAV-PHP.B-*NPCI* vs AAV9-*NPCI*."
3. To assess effect of treatment on behavior of NPC1 mice, gait analysis or maybe clasping test would have been appreciated. Could authors comment on why they decide to use their phenotype score and beam walk test, which seems not the best appropriated to the model.
 - We agree with the reviewer that both gait and hindlimb clasp can be very useful measures for NPC mouse models. In fact, both parameters are included in the composite phenotype score. For clarity, we have updated the results text to include the 6 measures of the composite score (p. 10, lines 164-165), along with descriptions in the Materials and Methods (p. 24, line 469) and legend for Figure 2 (p. 40, lines 869-870). We feel that the composite score – which additionally includes grooming, kyphosis, tremor, and ledge test – is a good readout measure that encompasses more of the overall phenotype. While acknowledging that objective quantitative data is preferred, we do not have access to specialized systems such as the Noldus CatWalk XT and hence the balance beam is a rapid, low cost alternative.
4. Regarding VGC determination; compare to the image shown in Figure 1, VGC for AAVPHPB is not so high in cerebrum, could authors detail the VGC in several brain area (hippocampus, cortex,...). And what about other peripheral organs, did VGC was assessed and why showing only liver?
 - All frozen samples collected from the study mice and subsequently used for gene copy number analysis have been homogenized. The brain was collected and homogenized in two parts: cerebellum and cerebrum (which includes areas such as hippocampus, cortex, and thalamus), thus brain sub-regions from these mice are not available.
 - Our experimental design involved collection of only brain and liver for gene copy number analysis, due to the large number of mice in the study. We selected these two organs because they both have a well-documented role in NPC disease, opting to further isolate the cerebellum due to the significant impact of Purkinje neuron loss on phenotypic manifestations.

5. In AAV9 treated mice, authors showed a number of 80VGC, no comments are done on a potential toxicity and it is difficult to really compare with AAVPHPB s the total injected dose was not the same.
 - We acknowledge that the doses for AAV9-*NPC1* and AAV-PHP.B-*NPC1* differ due to technical issues described in point 2, with AAV9 being administered at a higher dose. As mentioned above in point 2, this difference has been noted in the manuscript (p. 9, lines 138-139). While toxicity is possible with AAV9, we found no histological indicators consistent with hepatic injury or genotoxicity. Furthermore, the mice had no adverse clinical symptoms as have been described in NHPs or piglets treated with high doses of AAV9 early in life (Hinderer et al., 2018). Upon examination of H&E staining for liver in *Npc1*^{m1N/m1N} mice treated with AAV9-*NPC1*, the cytoarchitecture actually looked very similar to *Npc1*^{+/+} liver and showed no obvious signs of toxicity (Fig. 6 and Supplemental Fig. 8). Text describing these observations is now included in the results (p. 17-18, lines 331-338).
 - We have performed an additional analysis on the liver, quantification of the percent area positive for macrophages (Kuppfer cells) as assessed by CD68 labeling. No statistically significant differences were found between the AAV9-*NPC1* and AAV-PHP.B-*NPC1* treated mouse livers. In fact, the only significant difference was observed between *Npc1*^{m1N/m1N} mice administered saline and the *Npc1*^{+/+} mice, an expected finding.
 - Text has been added to p.16, lines 323-330: “Finally, the percentage of CD68⁺ area in sectioned liver tissue was compared between the different treatment groups (Supplemental Fig.7B). Livers from saline-injected *Npc1*^{m1N/m1N} mice displayed significantly larger CD68⁺ areas than did livers from *Npc1*^{+/+} mice (Kruskal-Wallis test with Dunn’s multiple comparison post-test; $P=.0109$). Although neither AAV9-*NPC1*- nor AAV-PHP.B-*NPC1*-treated *Npc1*^{m1N/m1N} mice showed significant differences from values for *Npc1*^{m1N/m1N} mice administered saline, there was a trend toward reduced CD68⁺ area in the AAV9-*NPC1*-treated mice which correlates with the assessments of vector copy number and NPC1 protein levels in liver.” Additionally, Supplemental Fig. 7B has been added with mean, standard deviation, and standard error of mean for the assessment. The reference Rodriguez-Gil et al, 2020 has been cited for method of quantification.
6. In figure 4, did expression of NPC1 in liver and cerebrum was evaluated in parallel as AAV9 liver display 80VGC and AAVPHPB cerebrum around 2-3 VgC however, the levels of expression seem similar, which is quite puzzling, Could the authors comment on that?
 - A couple of factors explain why the Western blot bands for NPC1 in liver and cerebrum appear similar. First, in order to detect NPC1 expression in the cerebrum, it was necessary to load 120 µg of protein vs. only 80 µg of protein for liver. Second, the acquisition settings for liver and cerebrum blots were configured independently to allow detection of bands within homogenate of either organ. A longer exposure time and higher gain were required to detect very faint bands in cerebrum (as evidenced by the over-exposed ladder in cerebrum blot). For liver, NPC1 expression was much stronger and required lower gain and exposure to visualize expression (again, lighter ladder indicates lower settings).
7. All the part regarding Ly6a receptor is quite disturbing for me, indeed it is known and published that at least in mice AAVPHPB need the Ly6a receptor to transduce the CNS, so why some mice in which My6 was not present where used for the study, a first screening of their mice expressing the Ly6 receptor should have been done to only use them for the study, could the authors comments on that? This for me is more supplementary data
 - Thank you for pointing out this important concern. Our study evaluating AAV-PHP.B in NPC disease commenced in 2016, but the first work documenting the Ly6a dependence of AAV-PHP.B in mice was not published until 2019 (Hordeaux et al, 2019), three years following the initiation of our study. Therefore, we were unaware of this confounding issue until all of our mice had already been injected and sacrificed. We have edited text on p. 12,

lines 215-217 to clarify this accordingly: “Following the completion of the AAV-PHP.B-*NPC1* experiments, other groups published work identifying strain-specific effects on the CNS transduction efficiency of AAV-PHP.B that are associated with different haplotypes at *Ly6a*...”

8. Finally, Iba1 staining is not convincing for me even in NPC1 saline mice, could authors presents a better staining for that
 - We appreciate the constructive feedback related to immunofluorescence labeling of IBA1. We have performed additional staining and quantification of the percent IBA1⁺ area relative to total region of interest. This data is included in Supplemental Fig. 7 and in the text on p. 16, lines 291-304: “Only modest improvement of gliosis in gene therapy treated mice compared to the saline-injected *Npc1*^{m1N/m1N} cohort was observed, as evidenced by microglial and astrocytic staining (anti-IBA1 and anti-GFAP, respectively). Quantification of microgliosis, or the percentage of IBA1⁺ area, was determined in lobules III, VI/VII, and IX of cerebellar tissue sections. Saline-injected *Npc1*^{m1N/m1N} mice displayed a significantly higher percentage IBA1⁺ area as compared to *Npc1*^{+/+} mice (Supplemental Fig. 7A; Kruskal-Wallis test with Dunn’s multiple comparison post-test, *P*=.0029 for lobule III, *P*=.0034 for lobules VI/VII, *P*=.0264 for lobule IX). IBA1⁺ area in specified lobules of cerebella from *Npc1*^{m1N/m1N} mice treated with either gene therapy vector were not significantly different from saline-injected *Npc1*^{m1N/m1N} or *Npc1*^{+/+} mice, though AAV9-PHP.B-*NPC1*-treated *Npc1*^{m1N/m1N} mice did trend toward greater reduction in pathology. This finding correlates with the higher AAV9-PHP.B-*NPC1* vector copy number and NPC1 protein levels in brain.”
9. For liver analysis, no comments on AAV9 toxicity related, and is not clear for me how a 80 VCG transduction would only need to a modest improvement. Could the authors comment on that
 - The revised manuscript now indicates that there were no signs of AAV9 toxicity, as described in point 5 above.
 - We agree that an average of 80 vector copies per cell seen in livers of AAV9-*NPC1* treated *Npc1*^{m1N/m1N} mice could suggest that liver pathology should be greatly reduced. As with any assay, there are limitations to the droplet digital PCR assay. In particular, two scenarios may account for higher copy numbers: a greater number of cells may have a lower copy number per cell *or* a smaller number of cells may have a higher copy number per cell. We cannot conclude which scenario best represents of our data.
10. For figure 2C and D, the normal growth curve which is present in supplementary seems more clear than the 2 extrapolate graphs
 - We agree that the weight curves are quite easy to visualize, however, they may be misinterpreted since mice continually drop out of the curve upon humane sacrifice. Therefore, weekly weights are not available for every mouse because of the different ages of sacrifice in the survival cohort. In contrast, the graph in 2C highlights when each mouse reached its peak weight. This represents a single time point per mouse and is available for every mouse in the study.
 - Graph 2D provides a single measurement of weight gain, loss, or maintenance for every mouse from 6-9 weeks of age, including *Npc1*^{+/+} mice. It is well documented that untreated *Npc1*^{m1N/m1N} mice experience a precipitous decline in weight during this time period. Successful therapeutic interventions often delay this phenotypic manifestation. Therefore, this measurement provides a quantifiable and easily visualized way to identify a positive impact.
 - For the reasons outlined above, we believe the weight curves should remain as supplemental figures.

August 4, 2021

RE: Life Science Alliance Manuscript #LSA-2021-01040RR

Dr. William J. Pavan
National Human Genome Research Institute
Genetic Disease Research Branch Building 49, Room 4A82 49 Convent Drive, MSC 4472
Bethesda, MD 20892-4472

Dear Dr. Pavan,

Thank you for submitting your Research Article entitled "Improved systemic AAV gene therapy with a neurotrophic capsid in Niemann-Pick disease type C1 mice". It is a pleasure to let you know that your manuscript is now accepted for publication in Life Science Alliance. Congratulations on this interesting work.

DISTRIBUTION OF MATERIALS:

Again, congratulations on a very nice paper. I hope you found the review process to be constructive and are pleased with how the manuscript was handled editorially. We look forward to future exciting submissions from your lab.

Sincerely,
